# Latent Representation Entropy Density for Distribution Shift Detection

**Fabio Arnez**[1]        **Daniel Alfonso Montoya Vasquez**[1]        **Ansgar Radermacher**[1]        **François Terrier**[1]

[1]Université Paris-Saclay, CEA, List, F-91120, Palaiseau, France

## Abstract

Distribution shift detection is paramount in safety-critical tasks that rely on Deep Neural Networks (DNNs). The detection task entails deriving a confidence score to assert whether a new input sample aligns with the training data distribution of the DNN model. While DNN predictive uncertainty offers an intuitive confidence measure, exploring uncertainty-based distribution shift detection with simple sample-based techniques has been relatively overlooked in recent years due to computational overhead and lower performance than plain post-hoc methods. This paper proposes using simple sample-based techniques for estimating uncertainty and employing the entropy density from intermediate representations to detect distribution shifts. We demonstrate the effectiveness of our method using standard benchmark datasets for out-of-distribution detection and across different common perception tasks with convolutional neural network architectures. Our scope extends beyond classification, encompassing image-level distribution shift detection for object detection and semantic segmentation tasks. Our results show that our method's performance is comparable to existing *State-of-the-Art* methods while being computationally faster and lighter than other Bayesian approaches, affirming its practical utility. Code is available at `https://github.com/CEA-LIST/LaREx`.

## 1 INTRODUCTION

As highly automated systems increasingly rely on DNNs to perform safety-critical tasks, confidence representation in DNN predictions has become crucial when deployed in the open world. Trustworthy DNN models should provide accurate predictions and detect samples that differ from those observed in the training distribution. Therefore, capturing information about *"what the model does not know"* is not only helpful but essential in safety-critical tasks and real-world deployment [Sun et al., 2021].

In image classification, multiple methods have been proposed for distribution shift detection by building DNN prediction confidence scores, among which post-hoc methods stand out mainly by their less-invasive nature and practical use [Yang et al., 2021, Ruff et al., 2021]. DNN predictive uncertainty offers a plain confidence representation. Existing Bayesian deep learning (BDL) methods provide a simple and principled approach to estimating DNN uncertainty. DNN predictive uncertainty with BDL methods has been used for detecting out-of-distribution (OoD) samples under the assumption that samples far away from the training distribution provide higher predictive uncertainty than samples observed in the training data [Ovadia et al., 2019, Kendall and Gal, 2017].

While BDL sampling-based methods are conceptually straightforward (*e.g.* , Monte-Carlo dropout), their practical implementation is hindered by substantial computational costs, limiting widespread adoption. Furthermore, recent works [Yang et al., 2021, Mukhoti et al., 2023] argue that BDL uncertainty is comparatively less effective for OoD detection when contrasted with more direct (deterministic) post-hoc methods. In addition, these problems can scale up to more complex computer vision tasks. In semantic segmentation, the lack of information on semantic structures and contexts yields miss-matches between anomaly pixel masses and pixel uncertainty regions [Di Biase et al., 2021, Xia et al., 2020]. In object detection, object distance and occlusion can impact the bounding-box predictive uncertainty for regression and classification [Feng et al., 2021, Wang et al., 2020]. Therefore, the limitations mentioned above lead to the open question: *Are DNN uncertainty-based confidence scores, with simple sample-based methods, still competitive for distribution shift detection?*

In this paper, we propose to use the uncertainty from intermediate latent representations (feature maps and embeddings) to detect distribution shifts at the image level. We leverage the latent representation entropy density from the training dataset and propose two new confidence scores (fully defined in Section 3.2) that we call `LaRED` & `LaREM` (`LaREx` for short). Our approach offers compelling benefits: 1) OoD data agnostic, *i.e.*, the score threshold is estimated only with in-distribution (InD) data; 2) simple post-hoc method that requires a single noise layer; 3) reduced runtime compared to sample-based BDL techniques and comparable to deterministic counterpart methods; 4) the presented scores can be applied to different CNN-based model architectures from different tasks. The paper contributions are summarized below:

1. We present two uncertainty-based confidence scores (`LaREx`) for image-level distribution shift detection that are computationally efficient compared with other BDL methods. We combine the benefits of simple sample-based methods for uncertainty estimation with density and distance-based methods for OoD detection.

2. We demonstrate the applicability of `LaREx` beyond image classification with more complex computer vision tasks, namely semantic segmentation and object detection tasks. Moreover, we show that image-level detection still has compelling benefits compared to more fine-grained detection schemes at the pixel or object level.

3. We performed extensive experimentation comparing the proposed confidence scores with standard baselines and benchmarks. In addition, we performed ablation studies presenting perspectives on enhancing the practical effectiveness of `LaREx` encompassing aspects such as regularization, dimensionality reduction, and the DNN layer to collect representations samples.

## 2 BACKGROUND

### 2.1 PROBLEM FORMULATION

Data distribution shift detection can be framed as a binary classification task. The classifier $\Omega$ aims at using a confidence score $\mathcal{S}$ with a corresponding threshold $\tau$ to determine (at inference time) whether a new input sample $x^*$ belongs to the training data distribution or not (OoD, anomalous samples), as presented in eq. (1):

$$\Omega\Big(\mathcal{S}(x^*),\tau\Big) \begin{cases} 1 & InD & \mathcal{S}(x^*) \geq \tau \\ 0 & OoD & \mathcal{S}(x^*) < \tau \end{cases} \quad (1)$$

Therefore, following the equation above, the goal is to derive a confidence score such that–*by convention in the literature*– positive InD samples have higher confidence scores and

vice versa for OoD or anomalous input samples. Then, the classifier $\Omega$ uses the confidence score $\mathcal{S}$ to get a notion of trust in the DNN and elicit its verdict.

### 2.2 RELATED WORK

In distribution shift detection, post-hoc methods aim to create confidence scores that have a minimal impact on the DNN architecture and the training process without altering the loss function. Post-hoc methods are presented below.

**Output-based Methods.** These methods aim at devising confidence scores based on the DNN outputs. Hendrycks and Gimpel [2016] proposed the first simple baseline method that uses the maximum softmax probability (MSP) as an InD membership score. Later work suggests using the maximum logit to outperform MSP [Hendrycks et al., 2019]. More recently, Liu et al. [2020b] proposed the energy score by summing up the prediction logits over all classes. In this line of work, ASH [Djurisic et al., 2022], DICE [Sun and Li, 2022], and ReAct [Sun et al., 2021] have worked on improving the energy score separability for InD and OoD data by modifying the activations of the penultimate layer and applying thresholding and scaling, sparsification, or clipping. In the context of uncertainty estimation, sample-based approximate Bayesian inference methods [Gal and Ghahramani, 2016, Lakshminarayanan et al., 2017] are used to generate multiple predictions for the same input sample, from which the predictive entropy and mutual information can be used as confidence scores [Kirsch et al., 2021, Mukhoti et al., 2023]. Unlike these methods, we do not use the DNN outputs to build our confidence scores.

**Density-based Methods.** These methods focus on modeling InD density using probabilistic models. In the context of discriminative models, deterministic uncertainty estimation methods Postels et al. [2020], Blum et al. [2021], Mukhoti et al. [2023] aim to estimate the embedding density while connecting to the traditional BDL approach. Another line of work employs generative models to represent the training data distribution, assuming that high-likelihood values correspond to InD samples and low-likelihood values to OoD samples. However, Nalisnick et al. [2018] showed that this assumption does not hold since the typical set of the data may not intersect with the high-likelihood region and adopt a typicality test approach using a batch of samples. Choi et al. [2018] suggests that OoD data may receive higher likelihoods due to epistemic errors and proposes using an ensemble of density models to address this issue. Follow-up work from Morningstar et al. [2021] propose assessing the typicality through multiple summary statistics from the model and their corresponding density estimates to build a score for a single sample. In contrast, our approach incorporates the ideas from both of the previous lines of work and uses the entropy density from intermediate representations to build our confidence scores.

**Distance-based Methods.** These methods assume that OoD samples reside in farther locations than InD samples from the training reference examples. Lee et al. [2018] proposed using the minimum Mahalanobis distance to all embedding centroids per class, assuming that the feature space follows a multivariate normal distribution. Recent work from Sun et al. [2022] shows promising results by following a non-parametric approach in the feature space and using the Kth nearest neighbor (KNN) distance. Other works Techapanurak et al. [2020], Nitsch et al. [2021] use the cosine similarity between class embeddings and test sample embeddings as a confidence score. Our proposed scores follow both the parametric and the non-parametric approach for entropy density estimation. The parametric version is used to compute the Mahalanobis distance.

**Detection in complex computer vision tasks.** For object detection, Du et al. [2022] proposed to modify the training procedure of an RCNN to synthesize virtual outliers in the feature space so that the energy score behaves differently for InD or OoD samples. More recently, Wilson et al. [2023] proposed training an auxiliary network to distinguish hidden state activations across the backbone of an RCNN for InD or OoD samples by generating outliers as corrupted images in the input space. In semantic segmentation, recent benchmarks Chan et al. [2021] present adapted common post-hoc methods for detecting anomalies at the pixel level despite the high execution runtime that hinders its practical utility. Instead, our approach proposes to detect shifts at the image level from a standpoint that is previous and complementary to signal the presence of potential finer anomalies. More on this discussion is found in Appendix F.

## 3 METHOD

We propose an uncertainty-based confidence score that leverages the entropy from an intermediate DNN latent representation. Taking inspiration from Morningstar et al. [2021], in our formulation, the DNN latent representation entropy is represented as a random variable $\Psi \sim f_\Psi(\psi)$, and we estimate its density by employing the InD training samples. Next, we use the estimated representation entropy density to build a confidence score that enables the detection of newly shifted samples (OoD samples). Below, we describe our approach to capture latent representation entropy, the InD entropy density $f_\Psi(\psi)$ estimation, and the confidence scores computation.

### 3.1 LATENT REPRESENTATIONS UNCERTAINTY

Key to our approach is the estimation of uncertainty from a DNN latent representation. A simple way to estimate uncertainty is by applying dropout [Srivastava et al., 2014] to add multiplicative noise to latent representation $\tilde{z}$, as presented in eq. (2):

$$z = \mathbf{m} \odot \tilde{z}, \;\; where \;\; \mathbf{m} \sim \mathcal{B}(p_m) \qquad (2)$$

where $\mathbf{m}$ is the vector of independent *Bernoulli* random variables—*the dropout mask*—and $p_m$ is the drop probability that has the same dimension as $\tilde{z}$. A vector $\boldsymbol{m}$ is sampled and multiplied element-wise with the latent code $\tilde{z}$ to produce a modified *"noisy"* latent code $z$, for which we would like to marginalize out the dropout mask noise as follows:

$$p_\theta(\mathbf{z} \mid \boldsymbol{x}) = \int p_\theta(\mathbf{z} \mid \boldsymbol{x}, \boldsymbol{m}) \underbrace{p(\boldsymbol{m}) \, d\boldsymbol{m}}_{\textit{dropout masks}} \qquad (3)$$

Thus, to get the uncertainty of the latent code **z**, we take multiple samples from $\mathbf{m}$ to generate multiple dropout masks so that we can produce a set of $M$ samples **z**, $\{z_i\}_{i=1}^M$ that approximate eq. (3). This set of samples, produced with a DNN with weights $\theta$ and input $x$, help us characterize the sampling distribution $p_\theta(\mathbf{z} \mid \boldsymbol{x})$, whose entropy is presented in eq. (4).

$$\mathbb{H}(\mathbf{z} \mid \boldsymbol{x}) = - \int p_\theta(\boldsymbol{z} \mid \boldsymbol{x}) \ln p_\theta(\boldsymbol{z} \mid \boldsymbol{x}) \, d\boldsymbol{z} \qquad (4)$$

From a practical point of view, we need a single dropout layer to get the samples $\{z_i\}_{i=1}^M$ to approximate the integral from eq. (3). In addition, during deployment, this situation allows us to speed up the sampling acquisition since we no longer need to pass an input sample throughout the whole DNN. We perform a single forward pass for a given input sample and capture the latent representation just before the target dropout layer. Then, we apply different dropout masks to the captured latent representation.

Our approach to capture the latent representation uncertainty is akin to the Monte-Carlo dropout (MCD) [Gal and Ghahramani, 2016] method for Bayesian approximation. However, our method differs since we apply dropout to produce multiple noisy versions of the representation. Thus, to distinguish with MCD, we use the term *z Monte-Carlo dropout (zMCD)* henceforth. We refer the reader to Appendix B for additional insights between zMCD and MCD from approximate Bayesian inference.

**zMCD on feature maps.** Standard dropout is ineffective when applied to convolutional neural networks (CNNs) since it does not remove semantic and spatial information from CNN feature maps. On the other hand, dropping continuous regions in 2D feature maps with *DropBlock* can help remove semantic information and enforce remaining units to learn features for the assigned task [Ghiasi et al., 2018]. This effect is also desired for capturing uncertainties to overcome the standard dropout limitation. Therefore, we follow the approach from Deepshikha et al. [2021] and use DropBlock to capture the uncertainty from feature maps.

**Feature map processing.** CNN feature maps are of the form $z \in \mathbb{R}^{C \times H \times W}$, where $C$, $H$ and $W$ denote the feature map number of channels, height, and width respectively.

We compute the mean of the feature map across the spatial dimensions ($H$ and $W$) so that the latent feature representation is reduced to a vector:

$$\boldsymbol{z}_{\mu_c} = \frac{1}{HW} \sum_{h=1}^{H} \sum_{w=1}^{W} \boldsymbol{z}(c, h, w), \text{ where } \boldsymbol{z}_{\mu_c} \in \mathbb{R}^C \quad (5)$$

## 3.2 REPRESENTATION ENTROPY DENSITY FOR DETECTING DISTRIBUTION SHIFTS

To start the entropy computation for detecting shifted samples, we first assume access to a training dataset $\mathcal{D}_t = \{\boldsymbol{x}_n, \boldsymbol{y}_n\}_{n=1}^N$ with $N$ samples. Now, we generate a set of zMCD samples $\{\boldsymbol{z}_i\}_{i=1}^M$ for each training sample $\boldsymbol{x}_n$ The resulting zMCD samples can then be used to approximate the entropy from eq. (4), using standard entropy estimators methods [Kozachenko and Leonenko, 1987]:

$$\hat{\mathbb{H}}_n\big(\{\boldsymbol{z}_i\}_{i=1}^M\big) \approx \mathbb{H}_n(\mathbf{z} \mid \boldsymbol{x}_n) \quad (6)$$

Consequently, we produce entropy estimation vector samples $\{\psi_n\}_{n=1}^N$ for the training dataset $\mathcal{D}_t$ (InD) samples:

$$\begin{aligned} \psi &= \hat{\mathbb{H}}(\mathbf{z} \mid \boldsymbol{x}) \\ \{\psi_n\}_{n=1}^N &= \hat{\mathbb{H}}_n(\mathbf{z} \mid \boldsymbol{x}_n), \forall \boldsymbol{x}_n \in \mathcal{D}_t \end{aligned} \quad (7)$$

The entropy estimation samples $\{\psi_n\}_{n=1}^N$ from $\mathcal{D}_t$ are used to estimate the InD entropy density function $f_\Psi \approx \hat{f}_\Psi$. $f_\Psi$ is estimated using Kernel Density Estimation (KDE), or we assume that $f_\Psi$ is a multivariate Normal distribution, parameterized by the estimated mean $\hat{\mu}_\psi$ and covariance $\hat{\Sigma}_\psi$ from $\{\psi_n\}_{n=1}^N$, as shown in eq. (8) and eq. (9) respectively.

$$\hat{f}_\Psi = \hat{f}_{KDE}\big(\{\psi_n\}_{n=1}^N\big) \quad (8)$$

$$\hat{f}_\Psi = \mathcal{N}\big(\hat{\mu}_\psi, \hat{\Sigma}_\psi\big) \quad (9)$$

At test or deployment time, we use the estimated InD entropy density $\hat{f}_\Psi$ to produce a confidence score for a new input sample $\boldsymbol{x}^*$. To this end, we produce a set of zMCD samples $\{\boldsymbol{z}_i^*\}_{i=1}^M$ to estimate the latent representation $\mathbf{z}^*$ entropy vector for a new input sample $\boldsymbol{x}^*$:

$$\psi_{\boldsymbol{x}^*} = \hat{\mathbb{H}}(\mathbf{z}^* \mid \boldsymbol{x}^*) \quad (10)$$

In the case of LaRED—*Latent Representation Entropy Density log-likelihood*—score, we compute the log-likelihood of the entropy estimation $\psi_{\boldsymbol{x}^*}$ for a new input sample $\boldsymbol{x}^*$, using the estimated entropy density function from eq. (8):

$$\text{LaRED}(\boldsymbol{x}^*) = \log \hat{f}_{KDE}\big(\psi_{\boldsymbol{x}^*}\big) \quad (11)$$

Equation (11) is equivalent to the confidence score from Morningstar et al. [2021]. However, in our confidence score, we use a single summary statistic instead of multiple summary statistics—*i.e.*, the latent representation entropy.

For the LaREM—*Latent Representation Entropy density Mahalanobis distance*—score, we compute the negative Mahalanobis distance, using the estimated density $\hat{f}_\Psi$ parameters from eq. (9) and the entropy estimation $\psi_{\boldsymbol{x}^*}$ for $\boldsymbol{x}^*$:

$$\text{LaREM}(\boldsymbol{x}^*) = -\Big(\big(\psi_{\boldsymbol{x}^*} - \hat{\mu}_\psi\big)^\top \hat{\Sigma}_\psi^{-1} \big(\psi_{\boldsymbol{x}^*} - \hat{\mu}_\psi\big)\Big) \quad (12)$$

Equation (12) is based on the score from Lee et al. [2018]. However, we do not perform per-class centroid distance computations. Moreover, the LaREM score uses negative distance values to align with the convention where InD samples have higher confidence score values.

**Entropy vector dimensionality reduction.** Following previous works [Lee et al., 2018, Postels et al., 2020, Yang et al., 2023], we apply principal components analysis (PCA) to reduce the dimensionality of the obtained entropy vectors $\psi_{\boldsymbol{x}^*}$. Entropy vectors have the same dimensions as the latent code $\boldsymbol{z}$ or $\boldsymbol{z}_{\mu_c}$. Thus, the goal is to reduce the dimensions from $C$ to $C'$ so that $\psi_{\boldsymbol{x}^*} \in \mathbb{R}^{C'}$, where $C' < C$. Applying PCA is particularly important for the LaRED score, given the common limitations of the KDE algorithm in high-dimensional spaces.

Figure 1 shows our approach to capturing uncertainty from latent representations (as described in Section 3.1) and presents an overview of both LaRED & LaREM confidence score setup and computation during deployment. LaRED & LaREM computation details are available in Appendix A.

## 4 EXPERIMENTS & RESULTS

The experimental evaluation in this section aims to answer the following questions: 1) How do LaRED & LaREM scores perform compared to other post-hoc baseline methods for distribution shift detection? 2) How do different design choices affect LaRED & LaREM performance? 3) Can LaRED & LaREM scale to more complex computer vision tasks with different DNN architectures?

**Evaluation Metrics.** We select three common metrics for detecting misclassified shifted (OoD and anomalous) samples to evaluate the proposed method. These metrics are: 1) **FPR95** measures the false positive rate (FPR) of OoD samples when the true positive rate (TPR) of InD samples is 95%; 2) the area under the receiving operating characteristic curve **AUROC**; and 3) the area under the precision-recall curve **AUPR**.

### 4.1 IMAGE CLASSIFICATION

**Experiment setup.** For the classification task we use a standard ResNet-18 [He et al., 2016] DNN trained with the CIFAR-10 (InD) dataset. For the OoD detection evaluation, we consider SVHN [Netzer et al., 2011], Places365 [Zhou et al., 2017], LSUN-Crop [Yu et al., 2015], LSUN-Resize [Yu et al., 2015], Textures [Cimpoi et al., 2014],

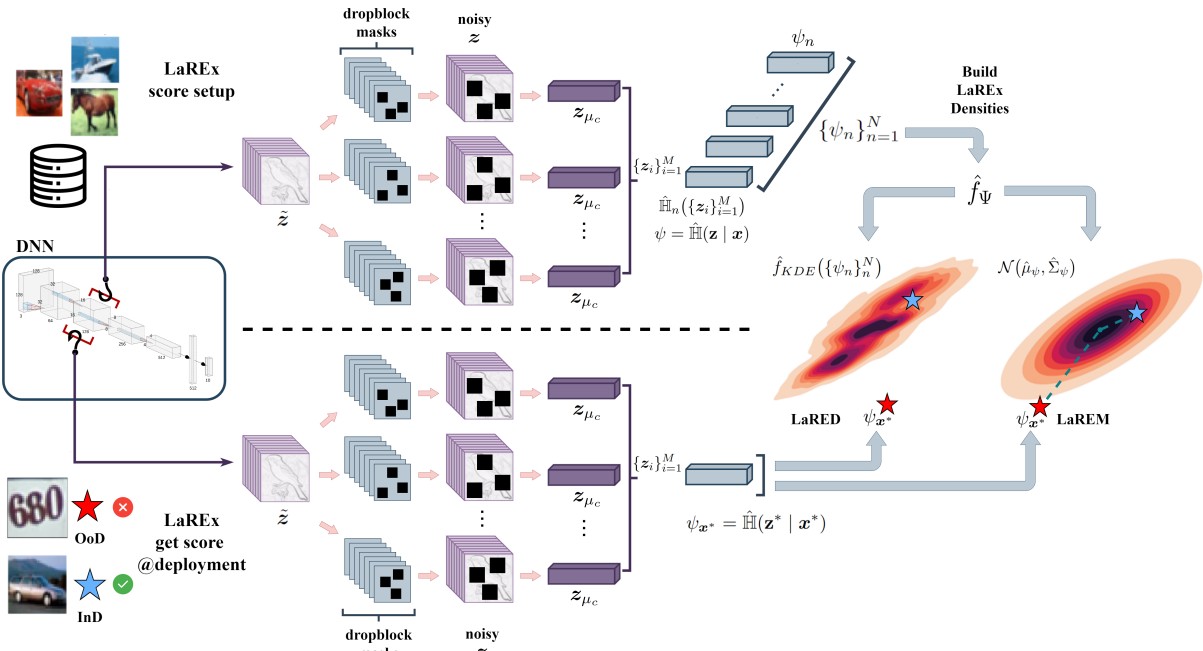

Figure 1: LaRED & LaREM confidence score overview. The left part of the figure shows the extraction of clean latent feature maps. In the center, multiple DropBlock masks are applied to the extracted feature map to get the uncertainty from the latent representation $p_\theta(\mathbf{z} \mid \boldsymbol{x})$. The upper part of the figure depicts the score setup computation to get the entropy density estimates $\hat{f}_\Psi$. The lower part of the figure shows the score computation during deployment.

iSUN [Xu et al., 2015], and Fashion MNIST [Xiao et al., 2017] datasets. We compare LaREx with common post-hoc detection methods from the literature that do not require additional OoD data. Uncertainty-based detection methods use 16 MC samples, *i.e.*, predictive entropy and predictive mutual information (MI), and LaREx (using MCD and zMCD, respectively). Additional experiment details are provided in Appendix C.

**Results.** Table 1 presents the average detection performance results over all the considered OoD datasets and the baselines described in Section 2.2. The results show that, in general, LaREx performance is on par with post-hoc baseline methods while being faster than other BDL approaches, as discussed in Section 4.4. In particular, LaRED & LaREM occupy the second and third positions, respectively, after KNN [Sun et al., 2022], denoting the benefits of not imposing a distributional assumption for the latent space. For both, LaRED & LaREM, the best results are obtained without applying PCA. Interestingly, methods that aim at improving the Energy score (React [Sun et al., 2021], DICE [Sun and Li, 2022], and ASH [Djurisic et al., 2022]) have worse performance than the vanilla Energy score [Liu et al., 2020b]. We believe that the drop in performance can be due to a sub-optimal parameter selection, *i.e.*, we used the best parameters proposed by the authors of each baseline without trying to find if other parameters performed better for this benchmark. Finally, for the other uncertainty-based methods (BDL w/MCD), the performance drop is more noticeable,

validating prior work [Kirsch et al., 2021, Mukhoti et al., 2023] observations.

**Where to collect zMCD samples?** To answer this question, we need to add a noise layer (DropBlock or dropout layer) at different locations of the neural network to enable zMCD sampling. To this end, we take into account the output of each residual block of the ResNet-18 as ideal places to take zMCD samples. We use DropBlock at the outputs of residual blocks 1 to 3 and a dropout layer for the output of the residual block 4. The dropout layer is used in the last position, given the dimensions of the embedding representation after the avg. pooling. Figure 2a shows the average detection performance when samples are taken using DropBlock or Dropout on different locations of the DNN described above. For both, LaRED & LaREM, the performance peaks when DropBlock is placed at the output of residual block-2 to take zMCD samples.

**DropBlock size matters.** Figure 2b presents the average detection performance across all OoD datasets for three different DropBlock sizes and a fixed drop probability of 0.5. For both methods, the performance is similar for block sizes of 3x3 and 5x5, with a subtle difference that favors a block size of 5x5 when inspecting the FPR95 results. A block size of 8x8 has a more noticeable drop in performance, affecting both methods. In this case, we attribute this effect to the fact that bigger DropBlock sizes tend to remove more relevant information, which can be vital for our methods.

Table 1: Image classification average detection performance results across seven OoD datasets. All the detection methods use the same DNN trained with CIFAR-10 (InD) dataset and with all the regularization options from Figure 3. The best results are shown in **bold**, second best are underlined.

| Method | FPR95 ↓ | AUROC ↑ | AUPR ↑ |
|---|---|---|---|
| MSP | 61.80 ± 9.60 | 84.44 ± 3.49 | 85.88 ± 3.80 |
| Pred. Entropy | 53.76 ± 16.98 | 88.44 ± 4.76 | 89.32 ± 4.96 |
| Pred. MI | 82.99 ± 9.17 | 77.29 ± 5.81 | 79.19 ± 5.38 |
| Energy | 47.96 ± 21.56 | 89.23 ± 7.18 | 89.27 ± 8.04 |
| ASH | 63.96 ± 18.21 | 80.87 ± 10.03 | 79.82 ± 12.52 |
| ReAct | 93.10 ± 2.58 | 53.76 ± 4.26 | 53.06 ± 4.77 |
| DICE | 81.51 ± 16.13 | 64.47 ± 12.10 | 62.19 ± 11.14 |
| DICE+ReAct | 92.31 ± 3.27 | 54.71 ± 5.73 | 54.27 ± 6.14 |
| KNN | **32.90 ± 20.30** | **92.65 ± 6.23** | **92.63 ± 6.32** |
| Mahalanobis | 57.35 ± 27.00 | 80.00 ± 11.45 | 78.70 ± 11.63 |
| LaRED(ours) | 33.16 ± 20.29 | 90.80 ± 6.50 | 90.80 ± 6.50 |
| LaREM(ours) | 37.33 ± 20.03 | 89.20 ± 6.62 | 87.60 ± 7.05 |

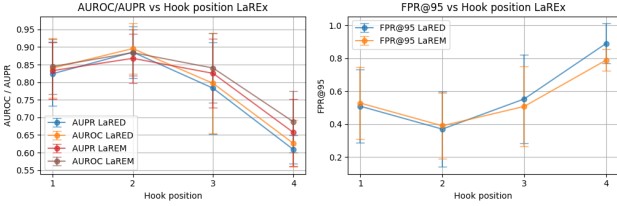

(a) DNN place for zMCD samples

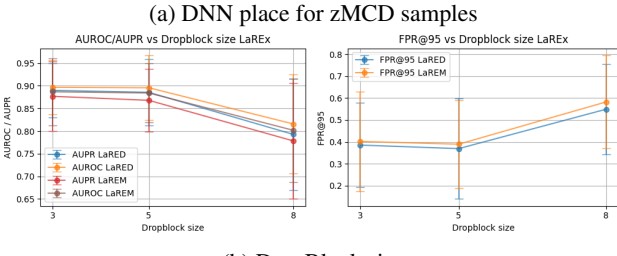

(b) DropBlock size

Figure 2: Impact of LaRED & LaREM design choices on average detection performance across all OoD data sets.

**Regularization improves performance.** We consider the impact of adding an extra dropout (DO) layer and data augmentation (DA) as simple ways to increase DNN regularization. The additional DO regularization layer is placed at the output of the ResNet encoder before the last linear layer. In addition, motivated by prior work on deterministic uncertainty estimation methods Mukhoti et al. [2023], Liu et al. [2020a], we also consider Spectral Normalization (SN) regularization. However, based on the work from Ghosh et al. [2020], we use SN only with the layers after the output of the residual block where we placed the DropBlock layer to regularize the latent space from where we take the zMCD samples. Figure 3 shows the average detection performance results across all the OoD datasets. In this figure, it is possible to observe that regularization impacts the performance of our method (and of baselines too). DA alone has a higher positive impact on performance compared to SN or DO. SN outperforms DO when applied alone. However, when DA is combined with DO, it outperforms DA+SN, validating

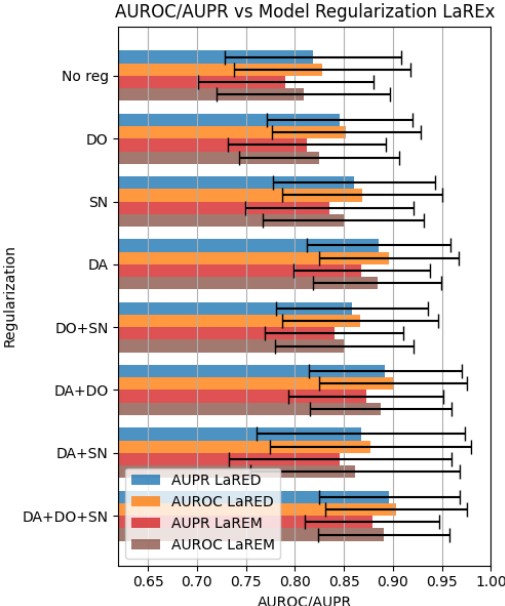

Figure 3: DNN regularization impact on LaRED & LaREM performance. DO: Dropout, SN: Spectral Normalization, DA: Data Augmentation.

the importance of noise injection during training. Moreover, applying all of them seems to be the most beneficial option for all OoD detection methods. We used a DNN trained with all the regularization options to compare our approach and the baseline methods from Table 1. We refer the reader to Appendix C for further details.

## 4.2 OBJECT DETECTION

**DNN model.** For the object detection task, we built on the work from Du et al. [2022] to detect distribution shifts with the Faster-RCNN architecture [Ren et al., 2015]. The main difference between our approach is that VOS [Du et al., 2022] aims to detect shifts at the object level. We, instead, aim to detect shifts at the image level. We use the Faster-RCNN vanilla pre-trained model from Du et al. [2022], trained on BDD-100K [Yu et al., 2020]. The models are implemented with the Detectron2 library [Girshick et al., 2018] with a ResNet-50 [He et al., 2016] backbone.

**Experiment setup.** We fine-tuned the trained model into two versions: the first one with a DropBlock layer and fine-tuned both the RPN and the RoI heads. To fine-tune and gather zMCD samples from the RPN, the DropBlock layer had a block size of 4 and a drop probability of 0.5 and was applied after the first convolutional layer. This layer outputs 256 feature maps of varying sizes (12x21, 24x42, 48x84, 96x168, and 192x336) for a total of 1280 feature maps. These feature maps are reduced by taking the mean of each of them as presented in eq. (5) to end up with simplified representation and entropy vectors of dimension 1280. In

the second version, we added the dropout layer after the penultimate layer of the network (the Box Head, BH) and fine-tuned only the RoI heads (Box Head and Box predictor). To fine-tune and capture zMCD samples from the box head, we used a drop probability of $0.5$. The output of this layer consists of a tensor of size 1000x1024 corresponding in the first dimension to the 1000 boxes highest ranked by the box pooler and the second dimension for the feature map for each of these boxes. To reduce the dimension of this feature map, we extracted the mean per box, obtaining an embedding of 1000 components. For the evaluation, we used as OoD the same splits provided by Du et al. [2022] of the MS-COCO [Lin et al., 2014] and OpenImages [Kuznetsova et al., 2020] datasets. Additional experiment details are provided in Appendix D from the supplementary material.

**Baseline Methods.** We implemented and adapted the following baselines for the object detection case: MSP [Hendrycks and Gimpel, 2016], Mutual information [Gal, 2016], Predictive entropy [Gal, 2016], energy score [Liu et al., 2020b], DICE [Sun and Li, 2022], ReAct [Sun et al., 2021], DICE+ReAct, and ASH [Djurisic et al., 2022]. For the energy-based methods, we implemented and evaluated two versions of each one: using the raw (R) output of the network (of 1000 results per image) and using the filtered (F) results after non-maximum suppression (NMS) (with variable size, typically of about 10-15 results per InD image). For MSP, pred. entropy, and mutual information, we took the output of the network after NMS. For ASH, we used the 80th percentile for pruning; for DICE, we used the 90th percentile for sparsifying; and for ReAct, we also used the 90th percentile for clipping. For SAFE [Wilson et al., 2023] and VOS [Du et al., 2022], we report the results from their respective papers.

**Results.** Table 2 summarizes the results for LaREx when using zMCD samples from the object detector RPN and the box head. For the version where the samples are collected from the RPN, LaRED w/40-PCA components and LaREM w/56-PCA components perform better than the version where samples are taken from the BH. In the latter, LaRED w/2-PCA components has better results, presumably thanks to the dimensionality of the latent representations. This agrees with our previous results for image classification, where placing the DropBlock layer at a more intermediate location in the DNN leads to better results than placing it closer to the output. In general, both LaRED & LaREM show a competitive performance compared to the other adapted baselines. In particular, LaRED shows the best AUROC results. Interestingly, our adapted F-ReAct, F-DICE, and F-ASH methods improve the F-Energy score detection performance results, and R-DICE shows the best results for OpenImages detection. Furthermore, despite not being a fair comparison, we report the results from VOS and SAFE showing that image level detection performance, in general, surpasses object level detection from recent works.

Table 2: Object detection OoD detection results. LaRED is applied at two different places of the Faster-RCNN DNN object detector trained with the BDD-100K dataset (InD dataset). The $^\dagger$ symbol denotes the 2nd DNN version that applies fine-tuning only to the RoI heads. The best results are in **bold** for each metric, and the second best are underlined. The ♣ symbol indicates the results as reported in Du et al. [2022], Wilson et al. [2023].

| Method | COCO | | OpenImages | |
|---|---|---|---|---|
| | FPR95 ↓ | AUROC ↑ | FPR95 ↓ | AUROC ↑ |
| R-Energy | 1.27±1.20 | 99.70±1.02 | 0.22±0.18 | 99.87±0.10 |
| F-Energy | 18.08±1.28 | 92.27±1.98 | 18.17±1.65 | 90.56±1.03 |
| R-ASH | 68.88±1.02 | 68.98±1.49 | 80.40±1.47 | 62.45±0.82 |
| F-ASH | 3.35±1.84 | 99.27±0.96 | 3.12±1.03 | 99.25±0.05 |
| R-React | 29.68±0.98 | 94.92±1.74 | 18.51±1.48 | 96.92±1.47 |
| F-React | 2.18±1.52 | 99.56±1.38 | 05.11±1.66 | 98.90±1.06 |
| R-DICE | 32.44±1.48 | 93.80±1.32 | **0.01±0.01** | **99.98±0.02** |
| F-DICE | 19.20±1.32 | 94.78±1.59 | 20.55±1.21 | 94.67±1.58 |
| R-DICE+ReAct | 24.94±1.74 | 94.36±1.46 | 97.55±1.16 | 52.14±1.98 |
| F-DICE+ReAct | 64.99±1.63 | 63.70±0.99 | 73.63±1.55 | 76.03±1.53 |
| MSP | **0.21±0.87** | 99.79±0.68 | 0.11±0.06 | 99.88±0.71 |
| Pred. Entropy | 68.88±1.02 | 68.98±1.49 | 80.40±1.47 | 62.45±0.82 |
| Pred. MI | 54.46±1.14 | 77.98±1.52 | 21.35±1.26 | 85.45±0.97 |
| LaRED RPN (ours) | 0.31±0.30 | **99.81±0.40** | 0.22±0.21 | 99.88±0.60 |
| LaREM RPN (ours) | 0.74±0.42 | 99.77±0.29 | 0.10±0.08 | 99.91±0.08 |
| LaRED BH $^\dagger$ (ours) | 12.07 ±0.60 | 97.48±0.80 | 10.33±1.20 | 97.54±0.90 |
| VOS-ResNet50♣ | 44.27±2.0 | 86.87±2.1 | 35.54±1.7 | 88.52±1.3 |
| SAFE-ResNet50♣ | 32.56±0.8 | 88.96±0.6 | 16.04±0.5 | 94.64±0.3 |

### 4.3  SEMANTIC SEGMENTATION

**DNN models.** For the semantic segmentation task, we concentrate on the application of the proposed method with the DeepLabv3+ [Chen et al., 2018], and U-Net [Ronneberger et al., 2015] architectures. To apply LaREx, we added a DropBlock layer at the output of both DeepLabv3+ and U-Net encoders using a block size of 8x8 and a drop probability of 0.5 to take zMCD samples. We train both DNNs with Cityscapes [Cordts et al., 2016] and Woodscape [Yogamani et al., 2019] datasets. DeepLabv3+ models $\mathcal{M}_1$ and $\mathcal{M}_2$, and U-Net models $\mathcal{M}_3$ and $\mathcal{M}_4$, respectively. Additional details are provided in Appendix E.

**Evaluation Datasets.** Motivated by Ahmed and Courville [2020] who argue that semantically similar samples are of practical relevance, we consider data with covariate shift for the experiments. When DNN is trained with Woodscape, we use Cityscapes data for the evaluation and vice-versa. Moreover, we consider the Failure Mode Effect Analysis in perception tasks from Ceccarelli and Secci [2022] and take into account InD data with perturbations and anomalies. Therefore, we include Cityscapes and Woodscape datasets with synthetic anomalies and the Woodscape-soiling [Yogamani et al., 2019] dataset for the evaluation.

**Baseline Methods.** In this task, we discarded post-hoc methods based on DNN outputs since we do not target pixel-level anomaly detection. Instead, we use the Mahalanobis [Lee et al., 2018] and KNN [Sun et al., 2022] distance as base-

Table 3: Semantic segmentation average distribution shift detection results for the evaluation datasets in DeepLabv3+ ($\mathcal{M}_1$, $\mathcal{M}_2$) and U-Net ($\mathcal{M}_3$, $\mathcal{M}_4$) architectures trained with Cityscapes ($\mathcal{M}_1$, $\mathcal{M}_3$) and Woodscape ($\mathcal{M}_2$, $\mathcal{M}_4$) datasets respectively. The best results are in **bold** for each metric.

| ID | Methods | Evaluation Datasets Average Performance | | |
|---|---|---|---|---|
| | | FPR95 ↓ | AUROC ↑ | AUPR ↑ |
| $\mathcal{M}_1$ | Mahalanobis | **1.07 ± 1.85** | **99.70 ± 0.46** | **99.75 ± 0.38** |
| | KNN | 8.28 ± 13.92 | 98.37 ± 2.38 | 98.53 ± 2.19 |
| | LaREM | 3.00 ± 5.20 | 99.39 ± 0.98 | 99.42 ± 0.93 |
| | LaRED-58 | 10.87 ± 18.60 | 97.04 ± 3.96 | 97.07 ± 4.16 |
| $\mathcal{M}_2$ | Mahalanobis | **1.59 ± 2.34** | **99.36 ± 0.48** | **99.58 ± 0.34** |
| | KNN | 7.96 ± 9.89 | 97.72 ± 1.36 | 96.09 ± 4.24 |
| | LaREM | 21.27 ± 13.73 | 93.57 ± 2.64 | 89.17 ± 5.40 |
| | LaRED-50 | 12.60 ± 8.06 | 96.24 ± 2.13 | 95.48 ± 3.01 |
| $\mathcal{M}_3$ | LaRED-50 | 17.79 ± 12.04 | 95.29 ± 3.84 | 95.02 ± 4.86 |
| $\mathcal{M}_4$ | LaRED-50 | 20.15 ± 14.61 | 95.42 ± 4.09 | 95.96 ± 4.05 |

Table 4: Deeplabv3+ uncertainty-based confidence scores runtime comparison on laptop PC with Intel i7-9750H CPU and NVIDIA RTX 2080. The best results are in **bold**.

| Method | Description | Runtime (ms) ↓ |
|---|---|---|
| Pred. Entropy | w/16 MCD samples | 416.01 ± 12.16 |
| LaREx | sampling only w/16 zMCD samples | 25.4 ± 3.22 |
| LaRED | score w/16 zMCD samples | **225.90 ± 7.00** |
| LaREM | score w/16 zMCD samples | 227.88 ± 8.91 |

line methods to compare `LaRED` & `LaREM`. However, these distance-based methods are different from those that target pixel-level anomaly detection [Chan et al., 2021]. Both methods use the representations from the penultimate layer. However, since the representations are now 2D feature maps, we take the mean of the feature maps as presented in eq. (5). Furthermore, for the Mahalanobis distance, we calculate a single entropy vector $\psi$ mean for the training (InD) set instead of dedicated means for each class.

**Results.** Table 3 presents the results for the models of both architectures for semantic segmentation. For both DeepLabv3+ models, `LaREM` w/o PCA has the best results, while `LaRED` w/58 PCA components and `LaRED` w/50 PCA components show the best results for the DNN models trained with Cityscapes and Woodscape, respectively. In all the models, `LaREx` performance is comparable to the other distance-based baselines. The reason for `LaREx` performance difference can be attributed to a sub-optimal selection of the parameters (*e.g.*, DropBlock location and size, PCA components) and to the presence of "clean" InD images in the evaluation datasets (in particular the case of Woodscape soiling dataset) that can be handled by the inherent robustness of the DNN. In contrast to the image classification case, for semantic segmentation, the Mahalanobis distance in both Deeplabv3+ models has the best performance results across the evaluated datasets, outperforming the KNN distance method. Presumably, the drop in KNN performance is due to the sub-optimal selection of the kth nearest neighbor. Note that we used the parameters proposed by the authors of each baseline, as in the image classification experiments. Next, in both U-Net models, `LaRED` w/50 PCA components show the best performance results. In general, the obtained results further validate the importance and effectiveness of the feature maps reduction by computing the mean to get a simplified representation as presented in eq. (5).

## 4.4 LAREX RUNTIME EXECUTION

The runtime results are presented in Table 4 for `LaRED` & `LaREM` and Predictive entropy with MCD-BDL in a DeepLabv3+ trained with Cityscapes. Although this can not be considered a completely fair comparison, given that MCD-BDL provides a pixel-level confidence score with the predicted uncertainty maps, the runtime results highlight the benefits of our approach in uncertainty-based distribution shift detection. For both, `LaRED` & `LaREM`, most of the computation time is dedicated to the score computation after the zMCD sampling step. `LaREx` sampling is faster since there is no need to perform a complete forward pass of the input samples as in the case of BDL-MCD. Nevertheless, most of `LaREx` runtime budget can presumably be attributed to data-transfer operations (GPU-RAM-CPU) for entropy estimation and the score computation.

## 4.5 ADDITIONAL INSIGHTS & DISCUSSION

**We found that LaREx can work without fine-tuning.** It is possible to just add "by hand" a dropout or DropBlock layer to take zMCD samples and get the method working. However, fine-tuning in most cases improves the results (see Appendix D). Interestingly, this situation puts in evidence an interesting direction for future work to connect theoretically `LaREx` with DICE and ASH.

**On LaREx limitations.** The most noticeable limitation is the need to perform sampling. However, it is not necessary to perform complete forward passes through the network for the method to work. As mentioned in Section 3.1, during deployment, we can speed up sampling by placing a hook on a desired DNN location to extract the feature representation. Then, with the added noise layer (DropBlock or dropout), we generate multiple noisy samples for the extracted representation. Another constraint lies in the absence of a predefined optimal location and size for DropBlock or dropout for any architecture to take the zMCD samples. Therefore, experimental iterations are required to find the best optimal location and parameters tailored to a specific DNN. However, empirically, we have found that DropBlock sizes of $\sim 20 - 40\%$ of the original feature map size and drop probabilities of $\sim 0.5$ are useful for capturing the desired variability in zMCD samples and lead to good results in multiple architectures.

**There is *no free lunch* in post-hoc methods.** Throughout our experiments, it became evident that each post-hoc method was influenced by stronger regularization and, notably, by data augmentation. Moreover, identifying a singular post-hoc method that universally outperforms others across diverse computer vision tasks proved challenging since the performance differences in the best detection methods are subtle. Consequently, an interesting line for future work involves the exploration of strategies to combine different confidence scores rather than relying solely on a single method for all tasks and their corresponding architectures.

## 5 CONCLUSION

We presented two uncertainty-based confidence scores, `LaREM` & `LaRED`, to detect data distribution shifts at the image level. The applicability of our confidence scores was demonstrated beyond simple classification, covering also the semantic segmentation and object detection tasks and the corresponding DNN architectures. Besides, our confidence score runtime achieves performance comparable to *SotA* methods while being faster than the traditional MCD method from the BDL framework, becoming an appealing uncertainty-based confidence score alternative. Finally, we provided additional insights into our method and extended the discussion by identifying and proposing different lines for future work.

## ACKNOWLEDGMENTS

This work has been supported by the French government under the "France 2030" program, as part of the SystemX Technological Research Institute within the *confiance.ai* Program (`www.confiance.ai`).

This publication was made possible by the use of FactoryIA supercomputer, financially supported by the Ile-de-France Regional Council.

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

# Latent Representation Entropy Density for Distribution Shift Detection (Supplementary Material)

**Fabio Arnez**[1]   **Daniel Alfonso Montoya Vasquez**[1]   **Ansgar Radermacher**[1]   **François Terrier**[1]

[1]Université Paris-Saclay, CEA, List, F-91120, Palaiseau, France

## A `LAREM` & `LARED` ALGORITHM

The computation details for `LaRED` & `LaREM` are available in Algorithm 1.

---

**Algorithm 1** Latent Representation Entropy Density-based Distribution Shift Detection: `LaRED` & `LaREM` Confidence Scores.

---

**Definitions:**
- Trained DNN $p_\theta(\mathbf{y} \mid \boldsymbol{x})$ with noise layer (dropout or dropblock) layer
- Feature extractor $p_\theta(\mathbf{z} \mid \boldsymbol{x})$ (Hook on Dropout or DropBlock layer)
- Training dataset samples $\mathcal{D}_t = \{\boldsymbol{x}_n, \boldsymbol{y}_n\}_n^N$

**procedure:** `setup_LaREx_score`:

    **for** each $\boldsymbol{x}_n \in \mathcal{D}_t$ **do**
        get $M$ zMCD samples $\{\boldsymbol{z}_i\}_{i=1}^M \sim p_\theta(\mathbf{z} \mid \boldsymbol{x}_n)$
        $\psi_n \leftarrow$ entropy $\left(\{\boldsymbol{z}_i\}_{i=1}^M\right)$
        save $\psi_n$ sample into $\Psi$
    **end for**

    $\Psi = \{\psi_n\}_n^N$

    **if** `LaRED` **then**
        $\hat{f}_\Psi = \hat{f}_{KDE}(\Psi)$
    **end if**
    **if** `LaREM` **then**
        $\hat{\mu}_\Psi \leftarrow mean(\Psi); \quad \hat{\Sigma}_\Psi \leftarrow covariance(\Psi)$
        $\hat{f}_\Psi = \mathcal{N}(\hat{\mu}_\Psi, \hat{\Sigma}_\Psi)$
    **end if**
**end procedure**

**function:** `get_LaREx_score`(*new sample* $\boldsymbol{x}^*$):

    get $M$ zMCD samples $\{\boldsymbol{z}_i\}_{i=1}^M \sim p_\theta(\mathbf{z} \mid \boldsymbol{x}^*)$
    $\psi_{x^*} \leftarrow$ entropy $\left(\{\boldsymbol{z}_i\}_{i=1}^M\right)$
    **if** `LaRED` **then**
        $\mathcal{S} = \log \hat{f}_{KDE}(\psi_{\boldsymbol{x}^*})$
    **end if**
    **if** `LaREM` **then**
        $\mathcal{S} = -\left(\left(\psi_{\boldsymbol{x}^*} - \hat{\mu}_\Psi\right)^\top \hat{\Sigma}_\Psi^{-1}\left(\psi_{\boldsymbol{x}^*} - \hat{\mu}_\Psi\right)\right)$
    **end if**
    **Return** $\mathcal{S}$
**end function**

---

Entropy estimation was implemented using the *Entropy-Estimators* library[1]. For the KDE, in all the experiments, we used a Gaussian kernel and bandwidth $= 1$, and the Scikit-Learn library[2].

# B   ZMCD RELATION WITH MCD FOR APPROXIMATE BAYESIAN INFERENCE

To capture uncertainty, we presented the *z Monte-Carlo Dropout* (zMCD) to produce noisy versions of a given layer latent representation. Although zMCD is similar to Monte-Carlo Dropout (MCD) for approximate Bayesian Inference in DNNs, it can not be considered part of the Bayesian deep learning family. First, we consider the Bayesian Neural Network (BNN) and its predictions described in Equation (13).

$$p(\mathbf{y}^* \mid \mathbf{x}^*, \mathcal{D}) = \int p(\mathbf{y}^* \mid \mathbf{x}^*, \boldsymbol{\theta}) \, p(\boldsymbol{\theta} \mid \mathcal{D}) \, d\boldsymbol{\theta} \tag{13}$$

To approximate the Equation (13), MCD performs a variational inference approximation to the intractable posterior of the wights $p(\boldsymbol{\theta} \mid \mathcal{D})$. In this case, dropout is still applied to the representation from a given layer, and it does not cancel the neural network weights by default.

Following the work from Gal and Ghahramani [2016], the neural network weight cancel to perform approximate Bayesian inference is achieved by arranging the next linear combination between the latent representation $\boldsymbol{z}$ with dropout or dropblock mask $\boldsymbol{m}$ and the layer weights $\boldsymbol{\theta}$, as shown below:

$$\begin{aligned}
\hat{\boldsymbol{y}} &= \sigma\Big( \big(\boldsymbol{z} \odot \boldsymbol{m}\big) \boldsymbol{\theta} + \boldsymbol{b} \Big) \\
\hat{\boldsymbol{y}} &= \sigma\Big( \boldsymbol{z} \, \big(\mathrm{diag}(\boldsymbol{m}) \cdot \boldsymbol{\theta}\big) + \boldsymbol{b} \Big)
\end{aligned} \tag{14}$$

In zMCD, we capture the latent (noisy) feature representation samples directly at the output of the DropBlock or Dropout layers. If we would like to turn our method into the Bayesian framework (MCD), we simply need to collect the activation samples at least after the next layer to respect the weight canceling presented in Equation (14).

# C   IMAGE CLASSIFICATION EXPERIMENTS

## C.1   DNN TRANING DETAILS

Architecture and training details can be found in Table 5. We used the Pytorch-lightning library for training and inference. The models with the lowest validation loss were saved and used for subsequent inference and OoD detection evaluation. Additionally, when we indicate that a model has spectral normalization (SN), we apply it after the position for the DropBlock layer. For example, for the models with a DropBlock layer after the second residual block, SN was applied in the 3rd and 4th residual blocks and the fully connected layer. Data augmentation methods used during training can be found in Table 5. The seed for all random generators was 9290.

Table 5: Image classification DNN training details

| | |
|---|---|
| Architecture | ResNet-18 |
| Epochs | 300 |
| Batch size | 64 |
| Image size | 128x128 |
| Loss | Focal |
| Optimizer | Adam |
| Optim. weight decay | $1 \times 10^{-4}$ |
| LR scheduler | Cosine annealing |
| LR scheduler $\eta_{min}$ | $1 \times 10^{-5}$ |

In Table 7, it is possible to find all the models we trained and tested for the OoD detection task with their corresponding validation set accuracy. We tested different DropBlock and Dropout layer locations. Models $\mathcal{M}_0$ to $\mathcal{M}_3$, vary the location

[1] https://github.com/paulbrodersen/entropy_estimators
[2] https://scikit-learn.org/stable/about.html

Table 6: Image classification DNN training: Data augmentation details

| Augmentation | Parameters |
|---|---|
| Random Crop | padding: img size / 8 |
| Random Color Jitter: | p=0.2 |
| • contrast | 10% |
| • brightness | 10% |
| • saturation | 10% |
| Random grayscale | $p = 0.1$ |
| Random vertical flip | $p = 0.3$ |
| Random affine: | $p = 0.2$ |
| • angle | 20° |
| • translation | 20% |
| • scale | 1% to 20% |

Table 7: All models trained and tested for image classification with CIFAR10 as InD

| ID | Dropblock | | | Dropout | | SN | DA | Val. Acc |
|---|---|---|---|---|---|---|---|---|
| | Loc. | Size | Prob. | Active | Prob. | | | |
| $\mathcal{M}_0$ | RSB1 | 10 | 0.4 | No | 0 | No | Yes | 89.4 |
| $\mathcal{M}_{1(-1-4)}$ | RSB2 | 5 | 0.4 | No | 0 | No | Yes | 89.2 |
| $\mathcal{M}_2$ | RSB3 | 3 | 0.4 | No | 0 | No | Yes | 88.2 |
| $\mathcal{M}_3$ | No | 0 | 0 | Yes | 0.3 | No | Yes | 88.7 |
| $\mathcal{M}_{1-2}$ | RSB2 | 8 | 0.4 | No | 0 | No | Yes | 88.8 |
| $\mathcal{M}_{1-3}$ | RSB2 | 3 | 0.4 | No | 0 | No | Yes | 89.2 |
| $\mathcal{M}_{1-1-1}$ | RSB2 | 5 | 0.4 | No | 0 | No | No | 84.2 |
| $\mathcal{M}_{1-1-2}$ | RSB2 | 5 | 0.4 | Yes | 0.3 | No | No | 84 |
| $\mathcal{M}_{1-1-3}$ | RSB2 | 5 | 0.4 | No | 0 | Yes | No | 86.3 |
| $\mathcal{M}_{1-1-4}$ | RSB2 | 5 | 0.4 | No | 0 | No | Yes | 89.2 |
| $\mathcal{M}_{1-1-5}$ | RSB2 | 5 | 0.4 | Yes | 0.3 | Yes | No | 86.9 |
| $\mathcal{M}_{1-1-6}$ | RSB2 | 5 | 0.4 | Yes | 0.3 | No | Yes | 88.3 |
| $\mathcal{M}_{1-1-7}$ | RSB2 | 5 | 0.4 | No | 0 | Yes | Yes | 90 |
| $\mathcal{M}_{1-1-8}$ | RSB2 | 5 | 0.4 | Yes | 0.3 | Yes | Yes | 89.7 |

after every residual block (RSB). After noting that the best location was after the second residual block ($\mathcal{M}_1$), we tested with different DropBlock sizes: 3, 5, and 8. These sizes were chosen due to the size of the feature maps, which for the second layer were of size $16 \times 16$. The best results were obtained with a DropBlock size of 5. Then, we fixed the DropBlock size and location, and we proceeded to test combinations of three different regularization techniques: Dropout, Spectral Normalization (SN), and data augmentation (DA), as described in Section 4.1. In Table 7, note that model $\mathcal{M}1$ and model $\mathcal{M}_{1-1-4}$ are exactly the same since they share the same parameters. Moreover, note that the nomenclature used in models from $\mathcal{M}_{1-1-1}$ to $\mathcal{M}_{1-1-8}$ correspond in the same order to the results presented in Figure 3. For example, $\mathcal{M}_{1-1-1}$ corresponds to "No reg." and $\mathcal{M}_{1-1-8}$ corresponds to "DA+DO+SN".

For the evaluation of all the methods, we took samples from the datasets. For CIFAR10 (InD), we took a random sample of 8400 images from the training set. The (InD) sampled images were used to build the entropy density estimation from both LaRED and LaREM scores. The same set of image samples was used to estimate the thresholds from DICE and ReAct and, in general, for all the InD score estimators for all the baselines. Then, the evaluation was performed using the test set of all OoD datasets and CIFAR10 (InD). In this case, we randomly sampled 5000 images from each OoD dataset to perform the evaluation of all baselines and LaREx. This was done with the intention of building balanced-sized datasets. Note that the Textures dataset already has a size of 5000 images. In Table 8, we compare the performance of using all the samples from the datasets and using the samples mentioned above, and we found that the differences are not substantial. Therefore, we proceeded to perform the evaluation with the set of samples.

## C.2 DETAILED RESULTS OOD DETECTION

Table 9 expands Table 7 and shows that the model with all the three additional regularization techniques obtains the best results. Furthermore, for model $\mathcal{M}_{1-1-8}$ (DA+DO+SN) we present the details of the results for all baselines and all datasets in Table 10 and Table 11. In both tables, it is possible to observe that KNN presents the best overall performance, and LaRED comes in second place in performance across all methods, validating the effectiveness of the non-parametric assumption.

Table 8: Image classification average OoD detection performance using the sampled datasets vs using the full datasets

| Datasets size | LaRED | | | LaREM | | |
|---|---|---|---|---|---|---|
| | AUPR ↑ | AUROC ↑ | FPR95 ↓ | AUPR↑ | AUROC↑ | FPR95↓ |
| Full datasets | 90.29 ± 6.76 | 91.02 ± 6.46 | 32.23 ± 19.95 | 88.22 ± 7.12 | 89.31 ± 6.59 | 36.43 ± 19.98 |
| Samples | 89.7 ± 6.72 | 90.80 ± 6.50 | 33.16 ± 20.29 | 87.60 ± 7.05 | 89.20 ± 6.62 | 37.33 ± 20.03 |

Table 9: LaREx results for all image classification models trained with CIFAR10 (InD) and all OoD datasets

| Model | LaRED | | | LaREM | | |
|---|---|---|---|---|---|---|
| | FPR95 ↓ | AUROC ↑ | AUPR ↑ | FPR95 ↓ | AUROC ↑ | AUPR ↑ |
| $\mathcal{M}_0$ | 50.80 ± 22.27 | 84.01 ± 8.45 | 82.40 ± 9.09 | 52.73 ± 21.95 | 84.45 ± 7.79 | 83.28 ± 7.99 |
| $\mathcal{M}_{1(-1-4)}$ | 36.89 ± 22.84 | 89.56 ± 7.13 | 88.55 ± 7.36 | 38.91 ± 20.09 | 88.41 ± 6.55 | 86.79 ± 6.93 |
| $\mathcal{M}_2$ | 55.25 ± 26.88 | 79.68 ± 14.18 | 78.30 ± 13.09 | 50.69 ± 24.46 | 84.03 ± 9.88 | 82.51 ± 9.82 |
| $\mathcal{M}_3$ | 89.03 ± 12.14 | 62.60 ± 6.60 | 60.86 ± 3.97 | 78.89 ± 6.50 | 68.75 ± 8.65 | 65.71 ± 9.51 |
| $\mathcal{M}_{1-2}$ | 54.89 ± 20.70 | 81.56 ± 10.97 | 79.26 ± 12.26 | 58.22 ± 21.21 | 80.17 ± 11.43 | 77.82 ± 12.74 |
| $\mathcal{M}_{1-3}$ | 38.51 ± 19.22 | 89.67 ± 5.93 | 88.98 ± 6.00 | 40.19 ± 22.75 | 88.67 ± 7.38 | **87.68 ± 7.63** |
| $\mathcal{M}_{1-1-1}$ | 56.89 ± 22.54 | 82.77 ± 9.04 | 81.83 ± 9.02 | 59.79 ± 21.78 | 80.08 ± 8.85 | 79.05 ± 8.95 |
| $\mathcal{M}_{1-1-2}$ | 53.07 ± 22.02 | 85.24 ± 7.57 | 84.54 ± 7.42 | 56.24 ± 21.20 | 82.44 ± 8.17 | 81.25 ± 8.05 |
| $\mathcal{M}_{1-1-3}$ | 45.88 ± 22.83 | 86.88 ± 8.12 | 86.02 ± 8.29 | 49.76 ± 21.31 | 84.94 ± 8.22 | 83.48 ± 8.57 |
| $\mathcal{M}_{1-1-4}$ | 36.89 ± 22.84 | 89.56 ± 7.13 | 88.55 ± 7.36 | 38.91 ± 20.09 | 88.41 ± 6.55 | 86.79 ± 6.93 |
| $\mathcal{M}_{1-1-5}$ | 46.78 ± 23.06 | 86.62 ± 7.95 | 85.82 ± 7.78 | 52.94 ± 20.68 | 85.06 ± 7.05 | 84.02 ± 7.04 |
| $\mathcal{M}_{1-1-6}$ | 36.67 ± 24.80 | 90.01 ± 7.54 | 89.18 ± 7.80 | 38.56 ± 22.26 | 88.71 ± 7.22 | 87.25 ± 7.91 |
| $\mathcal{M}_{1-1-7}$ | 40.61 ± 28.60 | 87.69 ± 10.22 | 86.75 ± 10.63 | 42.70 ± 27.64 | 86.11 ± 10.70 | 84.58 ± 11.35 |
| $\mathcal{M}_{1-1-8}$ | **33.16 ± 20.29** | **90.80 ± 6.50** | **89.70 ± 6.72** | **37.33 ± 20.03** | **89.20 ± 6.62** | 87.60 ± 7.05 |

Table 10: Detailed results for all methods for FMNIST, SVHN, Places and Textures OoD datasets

| Method | Fashion MNIST | | | SVHN | | | Places 365 | | | Textures | | |
|---|---|---|---|---|---|---|---|---|---|---|---|---|
| | FPR95↓ | AUROC↑ | AUPR↑ | FPR95↓ | AUROC↑ | AUPR↑ | FPR95↓ | AUROC↑ | AUPR↑ | FPR95↓ | AUROC↑ | AUPR↑ |
| MSP | 52.36 | 86.94 | 88.11 | 74.93 | 81.90 | 84.26 | 58.22 | 85.11 | 86.21 | 73.85 | 77.66 | 77.97 |
| Pred. entropy | 36.08 | 91.77 | 91.85 | 77.95 | 84.57 | 87.27 | 54.31 | 88.89 | 89.65 | 76.22 | 79.56 | 79.10 |
| MI | 86.20 | 78.63 | 80.36 | 94.10 | 69.82 | 72.71 | 78.75 | 79.78 | 81.48 | 86.29 | 73.93 | 74.98 |
| Energy | 25.58 | 94.71 | 95.11 | 85.86 | 76.06 | 77.42 | 47.26 | 90.70 | 91.10 | 66.63 | 82.67 | 78.11 |
| ASH | 40.86 | 91.33 | 91.52 | 90.40 | 67.00 | 68.11 | 68.45 | 79.58 | 77.87 | 73.95 | 68.85 | 58.93 |
| ReAct | 91.48 | 56.68 | 54.53 | 93.49 | 51.42 | 50.03 | 94.62 | 52.60 | 52.82 | 92.75 | 51.63 | 47.44 |
| DICE | 40.28 | 88.77 | 83.91 | 66.48 | 71.49 | 61.74 | 63.02 | 80.37 | 75.93 | 91.44 | 72.10 | 77.02 |
| DICE+ReAct | 89.96 | 65.22 | 66.00 | 86.90 | 60.34 | 58.21 | 94.98 | 52.15 | 52.86 | 95.34 | 50.33 | 46.87 |
| kNN | **21.48** | **95.24** | **95.21** | 78.31 | 78.59 | 78.40 | 26.30 | **94.85** | **94.73** | 25.37 | 94.64 | 93.91 |
| Mahalanobis | 78.40 | 73.28 | 74.31 | 84.27 | 65.16 | 63.16 | 36.38 | 90.22 | 88.54 | 8.65 | 97.79 | 96.80 |
| LaRED | 66.56 | 79.29 | 78.35 | **18.23** | **95.53** | **94.57** | **22.68** | 94.70 | 94.01 | **5.83** | **98.63** | **98.14** |
| LaREM | 66.10 | 79.35 | 77.76 | 24.66 | 93.71 | 92.27 | 30.02 | 92.34 | 91.02 | 7.54 | 98.09 | 97.24 |

Note that Table 1 is built based on the results from Tables 10 and 11. To better appreciate the performance of all OoD detection methods, we present the ROC curves of all baselines and `LaREx` for each OoD dataset in Figure 4. Finally, in Figure 5, we can find the density scores for `LaRED` across all OoD datasets. In such plots, we can appreciate the separation that `LaRED` achieves per OoD dataset.

Regarding the baseline methods, from the obtained results, we attribute the performance of ASH [Djurisic et al., 2022], ReAct [Sun et al., 2021], and DICE+ReAct [Sun and Li, 2022] to a sub-optimal choice of parameters. However, we employed the same optimal values found in the corresponding papers, *i.e.*, for ASH, we took the 80th percentile for pruning, for DICE the 90th percentile for sparsifying, and for ReAct, we also used the 90th percentile for clipping.

## C.3 TRAINING REGULARIZATION IMPACTS POST-HOC DETECTION METHODS

In addition to the previous results, we provide evidence for the claim that the performance metrics of all post-hoc methods were influenced by the regularization of the model during training. We statistically tested the hypothesis that the AUROC, the AUPR, and the FPR@95 were different for all baselines and `LaREx` across OoD datasets for two different models: one that was regularized with data augmentations, dropout, and spectral normalization and one that has none of these regularization techniques. These models correspond to $\mathcal{M}_{1-1-1}$ (No reg.) and $\mathcal{M}_{1-1-8}$ (DA+DO+SN) of Table 7 and Figure 3. Note that the training was in no way done with the goal of favoring any OoD detection in particular. Indeed, they are all post-hoc methods, so no specific training is needed for any of them. For this sake, having 12 OoD detection methods (all 10 baselines

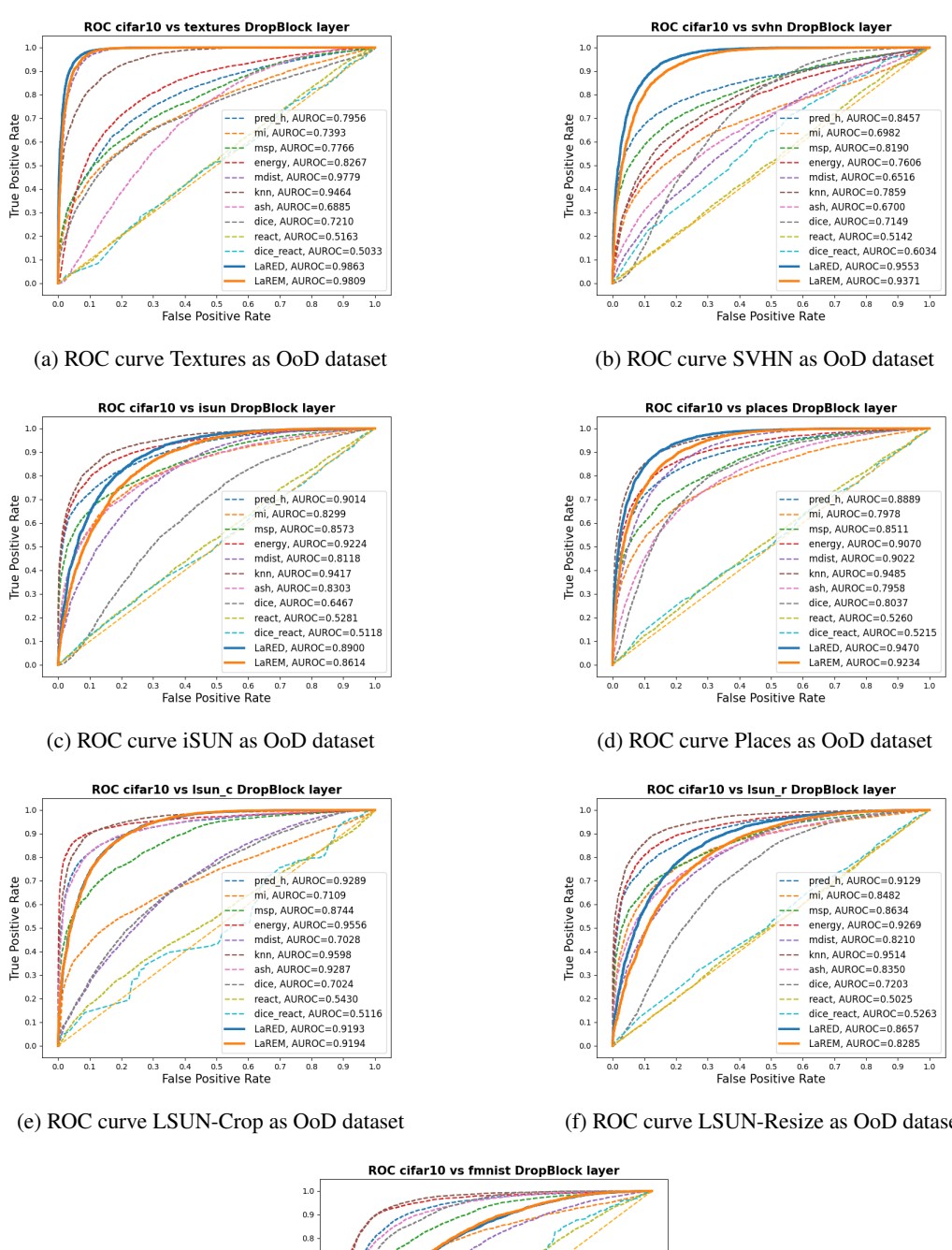

(a) ROC curve Textures as OoD dataset

(b) ROC curve SVHN as OoD dataset

(c) ROC curve iSUN as OoD dataset

(d) ROC curve Places as OoD dataset

(e) ROC curve LSUN-Crop as OoD dataset

(f) ROC curve LSUN-Resize as OoD dataset

(g) ROC curves Fashion-MNIST as OoD dataset

Figure 4: ROC curves for all OoD detection methods and all the OoD evaluation datasets using model $\mathcal{M}_{1\text{-}1\text{-}8}$ and CIFAR10 as InD. "mi": Predictive Mutual Information, "pred_h": Predictive Entropy, and "mdist": Mahalanobis distance

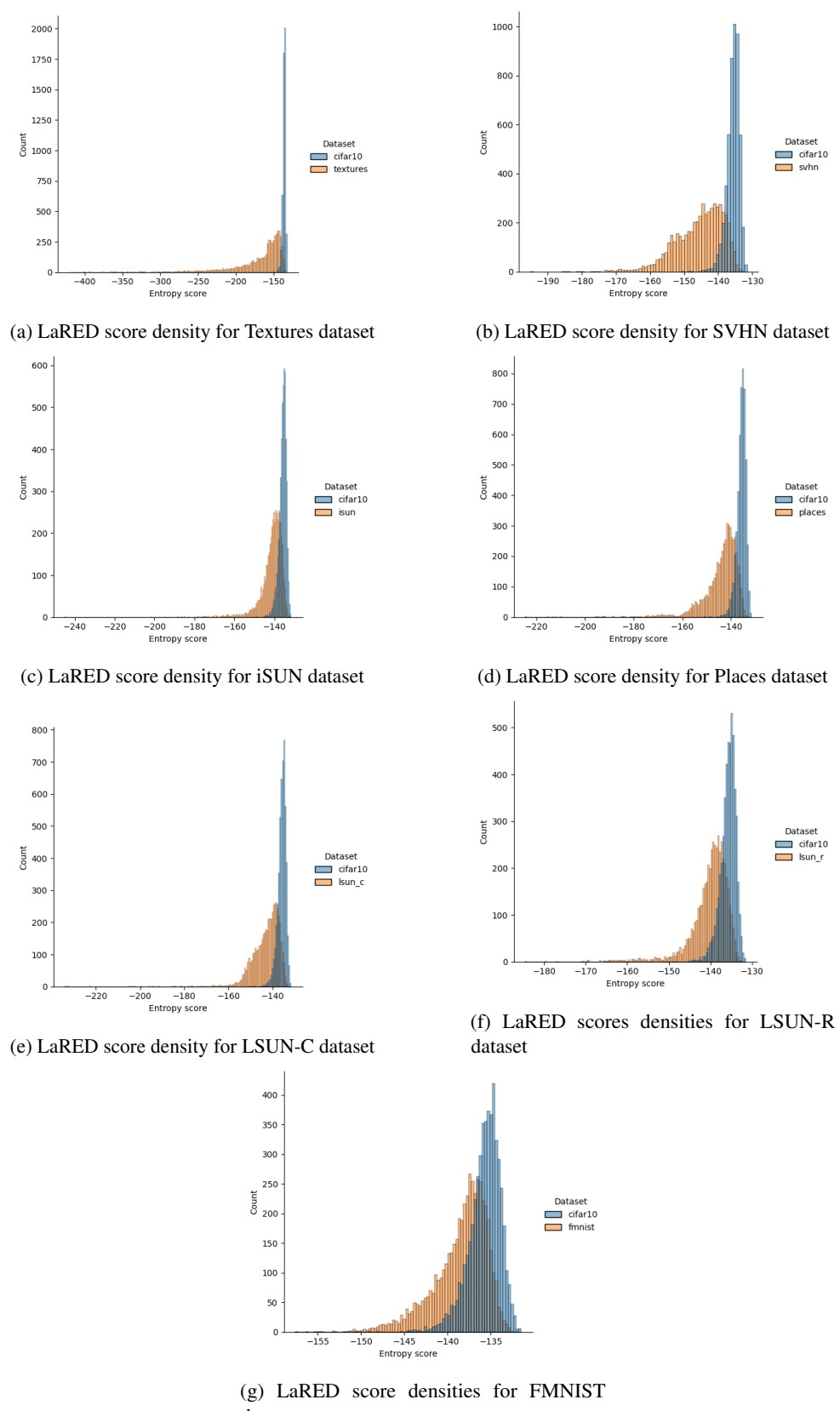

(a) LaRED score density for Textures dataset

(b) LaRED score density for SVHN dataset

(c) LaRED score density for iSUN dataset

(d) LaRED score density for Places dataset

(e) LaRED score density for LSUN-C dataset

(f) LaRED scores densities for LSUN-R dataset

(g) LaRED scores densities for FMNIST dataset

Figure 5: LaRED scores densities in model $\mathcal{M}_{1\text{-}1\text{-}8}$ for all OoD evaluation datasets

Table 11: Detailed results for all methods for LSUN-C, LSUN-R, and iSUN OoD datasets

| Method | LSUN-C | | | LSUN-R | | | iSUN | | |
|---|---|---|---|---|---|---|---|---|---|
| | FPR95↓ | AUROC↑ | AUPR↑ | FPR95↓ | AUROC↑ | AUPR↑ | FPR95↓ | AUROC↑ | AUPR↑ |
| MSP | 50.36 | 87.44 | 88.67 | 60.14 | 86.34 | 88.13 | 62.75 | 85.73 | 87.85 |
| Pred. entropy | 39.88 | 92.89 | 93.86 | 43.30 | 91.29 | 92.13 | 48.60 | 90.14 | 91.40 |
| Pred. MI | 92.22 | 71.09 | 73.96 | 70.68 | 84.82 | 86.36 | 72.69 | 82.99 | 84.54 |
| Energy | 27.68 | 95.56 | **96.52** | 39.70 | 92.69 | 93.46 | 43.02 | 92.24 | 93.23 |
| ASH | 39.22 | 92.87 | 93.40 | 66.62 | 83.50 | 84.67 | 68.27 | 83.03 | 84.27 |
| ReAct | 96.08 | 54.30 | 55.73 | 95.04 | 50.25 | 50.15 | 93.82 | 52.81 | 52.27 |
| DICE | 80.92 | 70.24 | 68.97 | 69.98 | 72.03 | 64.88 | 82.88 | 64.67 | 58.27 |
| DICE+ReAct | 90.24 | 51.16 | 51.77 | 94.02 | 52.63 | 52.81 | 94.76 | 51.18 | 51.42 |
| kNN | **21.24** | **95.98** | 96.15 | **26.38** | **95.14** | **95.50** | **31.25** | **94.17** | **94.57** |
| Mahalanobis | 77.18 | 70.28 | 67.76 | 58.94 | 82.10 | 81.31 | 57.66 | 81.18 | 79.08 |
| LaRED | 30.94 | 91.93 | 90.63 | 48.32 | 86.57 | 85.06 | 39.62 | 89.00 | 87.19 |
| LaREM | 30.56 | 91.94 | 90.70 | 56.04 | 82.85 | 80.47 | 46.40 | 86.14 | 83.76 |

Table 12: Shapiro-Wilk normality test results for OoD detection metrics for two models across baselines and across datasets

| Model | Metric | Statistic | p | Mean | Median | Std. |
|---|---|---|---|---|---|---|
| $\mathcal{M}_{1\text{-}1\text{-}8}$ | AUPR | 0.894 | $4.48 \times 10^{-6}$ | 79.03 | 83.83 | 14.96 |
| | FPR95 | 0.9373 | $4.94 \times 10^{-4}$ | 60.68 | 66.29 | 26.00 |
| | AUROC | 0.8976 | $6.16 \times 10^{-6}$ | 79.56 | 82.92 | 14.37 |
| $\mathcal{M}_{1\text{-}1\text{-}1}$ | AUPR | 0.9036 | $1.13 \times 10^{-5}$ | 78.39 | 81.71 | 10.65 |
| | FPR95 | 0.8936 | $4.18 \times 10^{-6}$ | 72.38 | 75.11 | 17.62 |
| | AUROC | 0.9542 | $4.63 \times 10^{-3}$ | 77.91 | 79.98 | 9.530 |

Table 13: Mann-Whitney U test results for OoD detection metrics for two models across baselines and across datasets

| Metric | Statistic | p |
|---|---|---|
| AUPR | 1.227 | 0.2195 |
| AUROC | 1.836 | 0.0662 |
| FPR@95 | -2.636 | 0.00838 |

plus `LaRED` and `LaREM`) and 7 OoD datasets, we have a sample of 84 data points per metric and model. We performed a normality test using the Shapiro-Wilk test, and the results from Table 12 show that it is not likely that the samples follow a normal distribution. Therefore, we proceeded to apply a non-parametric test: Mann-Whitney's U. The results of the test can be seen in Table 13.

From the results from Table 13 it follows that the models have statistically significant differences in their FPR@95, and near-significant results for the AUROC. The AUPR did not show statistically significant differences between the models. From this, together with the descriptive statistics in Table 12, it is possible to conclude that the regularization during training impacts all the tested OoD detection methods in a way that is statistically significant, in the sense that regularization seems to improve the performance of all tested methods. Also, these results are preliminary since they are not the main goal of our study, and further research is needed about how training procedures, regularization, and InD dataset characteristics may affect several OoD detection methods. We hypothesize that all of them have a general influence on the OoD detection task.

# D  OBJECT DETECTION EXPERIMENTS

## D.1  DNN TRANING DETAILS

For the object detection experiments, we took as a starting point the already trained model from Du et al. [2022], for a Faster-RCNN trained on BDD100K, using the *Detectron2* Girshick et al. [2018] library and openly available at their repository. Since it was not clear from the VOS paper or the code if the models were trained using Dropout, and clearly, they were not trained to use DropBlock, we ran experiments performing fine-tuning on the pre-trained models. For fine-tuning the RPN and Box Head, we froze the backbone and unfroze the RPN and subsequent layers completely, using a learning rate of $1 \times 10^{-4}$, and we kept the rest of the original hyper-parameters of the model (original learing rate was $2 \times 10^{-2}$). The used DropBlock size was 4, with a drop probability of 0.5. For fine-tuning the Box Heads, the backbone and the RPN were frozen, and the rest of the layers were trained with the same learning rates as described for the RPN. The dropout layer had a $p = 0.5$. All fine-tuning took place for 10 epochs. Moreover, we also tested the method by simply adding the Dropout or DropBlock layer (without fine-tuning) to the pre-trained network, and the results are shown in the respective section. We extracted 16 zMCD samples for all runs.

## D.2  DIMENSIONALITY REDUCTION

In this case of object detection, since the dimensionality of the extracted samples was high (a vector with 1280 components for the RPN samples and 1000 for the Box Head samples), we performed PCA before feeding the data to the final Kernel Density Estimator (KDE) or the Mahalanobis distance estimator. We used a randomly chosen sample of 8000 images from the training data from which we extracted the 16 zMCD samples per image, which makes up a tensor of dimension $(8000 \times 16, Z_s)$, where $Z_s$, the latent space size was either 1280 for the RPN or 1000 for the Box Head. The entropy was calculated, and we obtained a tensor of size $(8000, Z_s)$. We performed and evaluated the performance for several PCA dimension sizes: $\{1, 6, 14, 20, 24, 32, 40, 48, 56, 64, 72, 80\}$. The results of the PCA evaluation for the RPN hook models can be seen in Figure 6. For `LaRED`, there is a peak performance for 40 components, whereas for `LaREM`, the more components, the better.

## D.3  DETAILED RESULTS

For the detailed results of the experiments run on the number of PCA components to use, see Table 14. For all experiments, the random generators were seeded with the number 42. Based on the PCA results from Figure 6, we chose the 40 components for LaRED and 80 for LaREM. Furthermore, to visualize the separation achieved for our method, we present a PaCMAP [Wang et al., 2021] projection in 2 dimensions of the entropy scores in Figure 7.

Table 15, presents the results for fine-tuned vs not fine-tuned models. The former models have a better performance, and, in agreement with the Image classification, the results are better for the RPN, which is a more intermediate layer. Furthermore, Table 16 presents the impact of the number of zMCD samples in the performance metrics. In general, we observe that the more samples that are taken, the better. However, even with 5 zMCD samples, the performance drop is not extreme, which shows the robustness of the presented method and the possibility of experimenting and finding a trade-off with fewer zMCD samples, which indeed is one of the main limitations of our approach. Finally, Figure 8 presents a visualization of the performance from the detection methods in terms of ROC curves. We observe that most methods achieve high performance, validating the separability from both classes (InD and OoD) depicted previously in Figure 7.

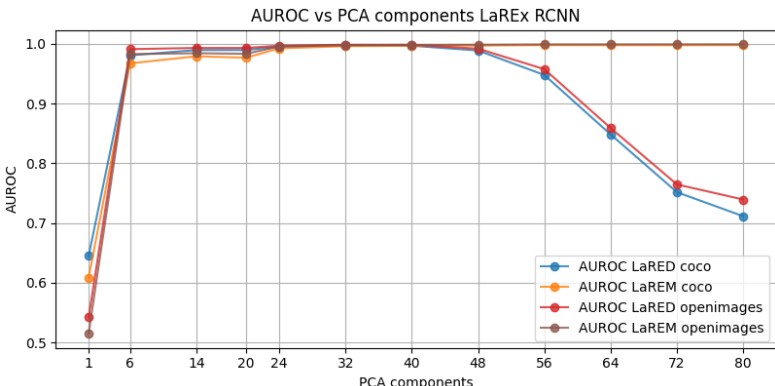

Figure 6: `LaRED` & `LaREM` evaluation of AUROC for several number of PCA components

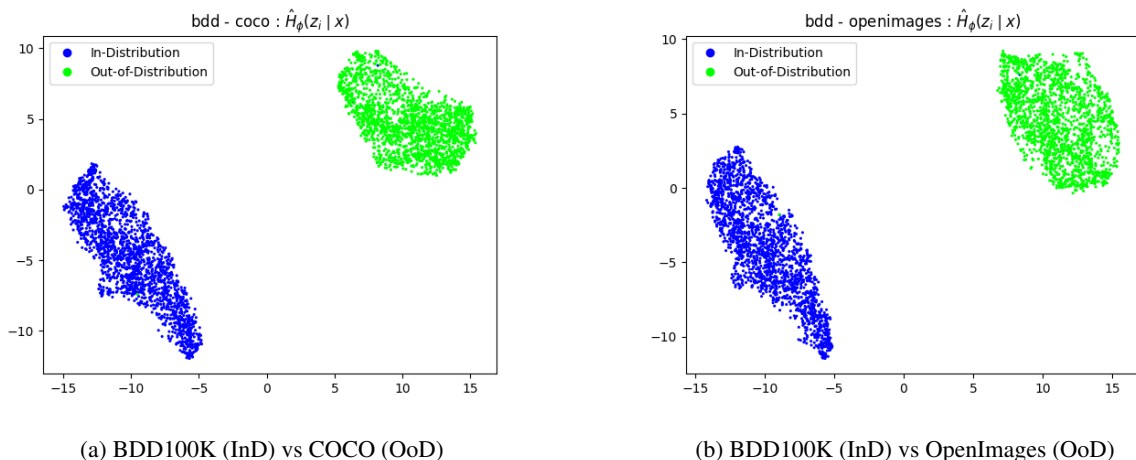

(a) BDD100K (InD) vs COCO (OoD)

(b) BDD100K (InD) vs OpenImages (OoD)

Figure 7: Faster R-CNN BDD-100K (InD): Entropy vectors 2D projection comparison using PaCMAP

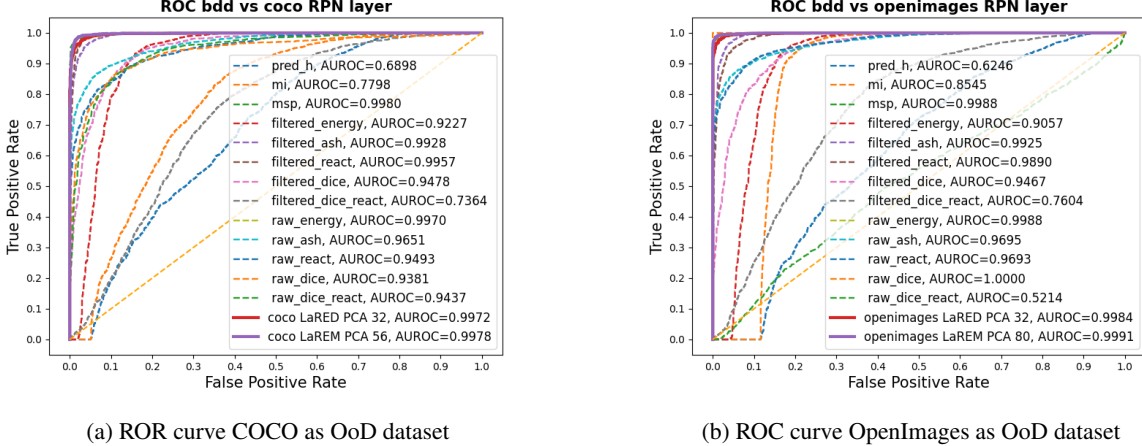

(a) ROR curve COCO as OoD dataset

(b) ROC curve OpenImages as OoD dataset

Figure 8: Object Detection with Faster-RCNN: ROC curves for all OoD detection methods for COCO and OpenImages as OoD datasets, fine-tuned model, LaRED RPN. "mi": Predictive Mutual Information, "pred_h": Predictive Entropy

Table 14: Detailed results for number of PCA components LaREx Object detection InD BDD100k, fine-tuned model

| Model | PCA components | AUPR COCO | AUPR OpenImages | AUROC COCO | AUROC OpenImages | FPR95 COCO | FPR95 OpenImages |
|---|---|---|---|---|---|---|---|
| | 1 | 63.7 | 55.05 | 64.57 | 53.99 | 93.03 | 93.41 |
| | 6 | 98.2 | 99.23 | 98.06 | 99.07 | 8.98 | 3.8 |
| | 14 | 99.34 | 99.56 | 99.21 | 99.44 | 3.35 | 1.98 |
| | 20 | 99.26 | 99.41 | 98.08 | 99.26 | 3.85 | 2.72 |
| | 24 | 99.51 | 99.58 | 99.43 | 99.5 | 2.5 | 1.93 |
| LaRED | 32 | 99.78 | 99.89 | 99.75 | 99.88 | 1.01 | 0.45 |
| RPN | 40 | **99.84** | **99.89** | **99.81** | **99.87** | **0.32** | **0.22** |
| | 48 | 99.26 | 99.36 | 99.01 | 99.1 | 1.49 | 1.02 |
| | 56 | 96.36 | 96.8 | 95.11 | 95.55 | 42.76 | 38.78 |
| | 64 | 87.43 | 89 | 84.82 | 85.89 | 49.57 | 50.48 |
| | 72 | 77.22 | 79.86 | 75.16 | 76.46 | 48.88 | 48.89 |
| | 80 | 73.34 | 77.25 | 71.14 | 73.93 | 50.9 | 48.32 |
| | 1 | 60.62 | 53.95 | 60.88 | 51.55 | 93.29 | 93.01 |
| | 6 | 96.89 | 98.51 | 96.72 | 98.26 | 16.38 | 9.02 |
| | 14 | 98.27 | 98.75 | 97.9 | 98.41 | 10.15 | 7.55 |
| | 20 | 98.11 | 98.7 | 97.66 | 98.32 | 11.91 | 8.8 |
| | 24 | 99.36 | 99.62 | 99.22 | 99.52 | 3.29 | 1.53 |
| LaREM | 32 | 99.64 | 99.8 | 99.59 | 99.76 | 1.64 | 0.85 |
| RPN | 40 | 99.7 | 99.85 | 99.66 | 99.82 | 1.48 | 0.68 |
| | 48 | 99.73 | 99.87 | 99.69 | 99.84 | 1.11 | 0.34 |
| | 56 | 99.79 | 99.91 | 99.77 | 99.89 | 0.74 | 0.17 |
| | 64 | 99.78 | 99.92 | 99.75 | 99.9 | 0.85 | 0.11 |
| | 72 | 99.78 | 99.92 | 99.75 | 99.9 | 0.9 | 0.056 |
| | 80 | **99.8** | **99.93** | **99.76** | **99.91** | **0.79** | **0.012** |

Table 15: Detailed results for fine-tuned vs not fine-tuned models LaREx Object detection InD BDD100k

| Methods | OoD: COCO | | OoD - OpenImages | | mAP |
|---|---|---|---|---|---|
| | FPR95 | AUROC | FPR95 | AUROC | |
| LaRED RPN Fine-tuned | **0.31±0.3** | **99.81±0.4** | 0.22±0.21 | 99.88±0.6 | 28.0 |
| LaREM RPN Fine-tuned | 0.79±0.58 | 99.77±0.26 | **0.11±0.09** | **99.91±0.08** | 28.0 |
| LaRED RPN No Fine-tune | 0.90±0.8 | 99.79±0.3 | 0.22±0.4 | 99.89±0.7 | 31.21 |
| LaRED FC Fine-tuned | 12.07±0.6 | 97.48±0.8 | 10.33±1.2 | 97.54±0.9 | 29.6 |
| LaRED FC No Fine-tune | 42.12±0.8 | 90.50±0.7 | 31.51±0.7 | 93.20±0.5 | 31.21 |

Table 16: Detailed results for the number of zMCD samples to take LaRED RPN Object detection InD BDD100k, fine-tuned model

| zMCD | AUPR COCO | AUPR OpenImages | AUROC COCO | AUROC OpenImages | FPR95 COCO | FPR95 OpenImages |
|---|---|---|---|---|---|---|
| 20 | **99.84** | **99.89** | **99.81** | **99.87** | **0.31** | **0.22** |
| 16 | 99.71 | 99.87 | 99.67 | 99.83 | 1.48 | 0.34 |
| 10 | 99.19 | 99.57 | 99.07 | 99.46 | 4.25 | 2.04 |
| 5 | 96.45 | 97.41 | 96.1 | 97.02 | 20.58 | 16.41 |

# E    SEMANTIC SEGMENTATION EXPERIMENTS

## E.1    DNN TRAINING DETAILS

For the semantic segmentation task, we consider the DeepLabv3+ [Chen et al., 2018], and U-Net [Ronneberger et al., 2015] architectures and the Cityscapes and Woodscape datasets for training (InD). In the Deeplabv3+ architecture[3], we added a DropBlock layer at the output of the ResNet encoder using a block size of $8 \times 8$ and drop probability $p = 0.5$ to take zMCD samples. The encoder output results in a tensor of shape $W/16 \times H/16 \times 2048$, where $W$ and $H$ represent the input image width and height, respectively, and the last dimension corresponds to the number of channels. For the U-Net architecture, we place a DropBlock layer at the output of the encoder using a block size of $8 \times 8$ and drop probability $p = 0.5$ to take zMCD samples. For the U-Net DNN trained with the Woodscape dataset, the encoder output has 128 channels. The U-Net DNN trained with the Cityscapes dataset has an encoder output with 256 channels. Table 17 summarize the used DNN training hyperparameters for each architecture and dataset.

## E.2    EVALUATION DATASETS

As mentioned in Section 4.3, we consider data with covariate shift for the semantic segmentation experiments. We used the Albumentations[4] library to create a *synthetic anomalies* version of the InD datasets. For the synthetic anomalies, we used the *Random Fog* and *Random Sun flare* transforms, and we implemented a custom transform to add the *Mud on lens* effect. Figure 9 shows samples on the (InD) training sets, while Figures 10 to 12 show samples of the datasets with covariate-shift used for evaluation.

## E.3    SEMANTIC SEGMENTATION DETAILED RESULTS

We use all the training dataset samples to set up and compute the InD scores from `LaREx` and the implemented baselines. the evaluation is implemented using all the samples from the validation and test sets. Tables 19 and 20 present the detailed performance results for each evaluated distribution shift dataset, in Deeplabv3+ and U-Net, respectively. The reason for `LaREx` performance difference can be attributed to a sub-optimal selection of the parameters and to the presence of "clean" InD images in the evaluation datasets. In contrast to the other experiments, the Mahalanobis distance in both Deeplabv3+ models has the best performance results across the evaluated datasets. This was also the case for `LaREM` when compared with `LaRED`. We attribute the dominance of the Mahalanobis-based methods to entropy vector dimensionality since no dimensionality reduction (w/PCA) is performed that might suppress useful information for the detection. The entropy vectors 2D projection using PaCMAP [Wang et al., 2021] are displayed in Figures 13 and 14 for Deeplabv3+, and in Figures 19 and 20 for U-Net, validating the effectiveness of the entropy vectors for the distribution shift detection task, and supporting our analysis of the results. Moreover, Figures 15 to 18 show the `LaREx` score comparison for each evaluated dataset in Deeplabv3+, and Figures 21 and 22 show the `LaRED` score comparison for each evaluated dataset in U-Net.

---

[3]`https://github.com/VainF/DeepLabV3Plus-Pytorch`
[4]`https://albumentations.ai/`

Table 17: Semantic Segmentation DNN training details

| Parameter | Deeplabv3+ | | U-Net | |
|---|---|---|---|---|
| | Cityscapes | Woodscape | Cityscapes | Woodscape |
| img size | 512x256 | 640x483 | 128x256 | 128x256 |
| epochs | 1500 | 350 | 1800 | 1400 |
| batch size | 16 | 8 | 16 | 16 |
| Loss | Focal | Focal | CE | CE |
| Optim | SGD | SGD | Adam | Adam |
| Weight decay | $5 \times 10^{-4}$ | $5 \times 10^{-4}$ | - | - |
| LR scheduler | Cosine annealing | Cosine annealing | Cosine annealing | Cosine annealing |
| LR scheduler $\eta_{min}$ | $1 \times 10^{-3}$ | $1 \times 10^{-3}$ | $2.3 \times 10^{-5}$ | $2.3 \times 10^{-5}$ |

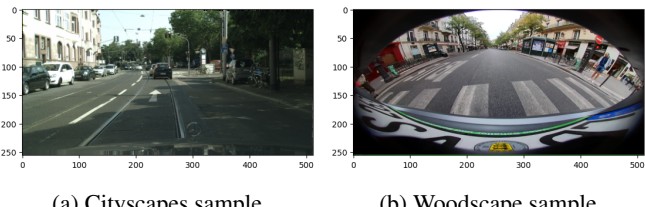

(a) Cityscapes sample      (b) Woodscape sample

Figure 9: Semantic segmentation DNN InD (training) datasets samples

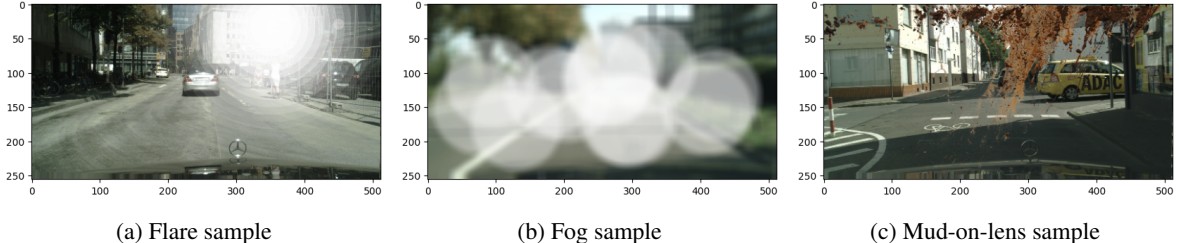

(a) Flare sample      (b) Fog sample      (c) Mud-on-lens sample

Figure 10: Cityscapes-Anomalies dataset samples

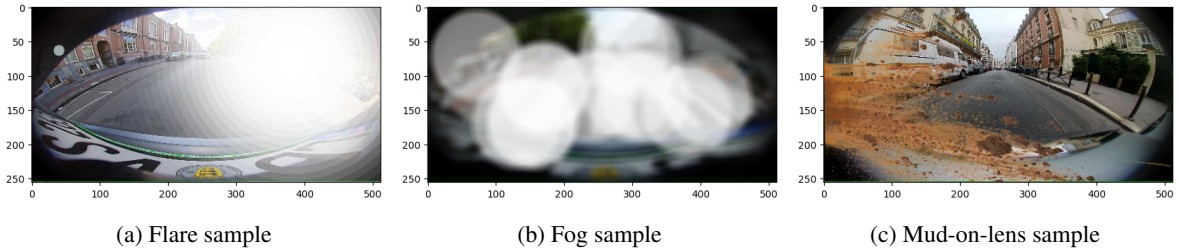

(a) Flare sample      (b) Fog sample      (c) Mud-on-lens sample

Figure 11: Woodscape-Anomalies dataset samples

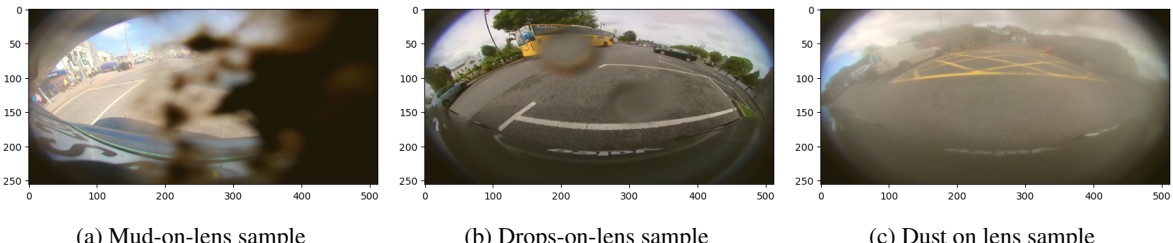

(a) Mud-on-lens sample      (b) Drops-on-lens sample      (c) Dust on lens sample

Figure 12: Woodscape-Soiling dataset samples

| Methods | Cityscapes-Anomalies | | | Woodscape | | | Woodscape-Soiling | | |
|---|---|---|---|---|---|---|---|---|---|
| | FPR95 ↓ | AUROC ↑ | AUPR ↑ | FPR95 ↓ | AUROC ↑ | AUPR ↑ | FPR95 ↓ | AUROC ↑ | AUPR ↑ |
| Mahalanobis | 3.21 | 99.17 | 99.31 | 0.0 | 99.94 | 99.95 | 0.0 | 99.98 | 99.99 |
| KNN | 24.35 | 95.62 | 96.0 | 0.48 | 99.62 | 99.67 | 0.0 | 99.87 | 99.92 |
| LaREM-2048 | 9.0 | 98.26 | 98.34 | 0.0 | 99.91 | 99.91 | 0.0 | 99.99 | 100.0 |
| LaRED-PCA58 | 32.35 | 92.48 | 92.27 | 0.26 | 99.14 | 99.28 | 0.0 | 99.51 | 99.65 |

Table 18: Deeplabv3+ trained w/Cityscapes dataset: distribution shift detection results

| Methods | Woodscape-Anomalies | | | Cityscapes | | | Woodscape-Soiling | | |
|---|---|---|---|---|---|---|---|---|---|
| | FPR95 ↓ | AUROC ↑ | AUPR ↑ | FPR95 ↓ | AUROC ↑ | AUPR ↑ | FPR95 ↓ | AUROC ↑ | AUPR ↑ |
| Mahalanobis | 0.48 | 99.6 | 99.71 | 0.0 | 99.67 | 99.84 | 4.28 | 98.8 | 99.2 |
| KNN | 4.51 | 98.69 | 98.96 | 0.0 | 99.79 | 99.89 | 10.44 | 97.93 | 98.41 |
| LaREM-2048 | 29.72 | 92.78 | 88.5 | 5.43 | 96.52 | 94.88 | 28.67 | 81.42 | 84.14 |
| LaRED-PCA50 | 20.39 | 94.32 | 92.48 | 4.3 | 98.54 | 98.5 | 13.11 | 95.87 | 95.45 |

Table 19: Deeplabv3+ trained w/Woodscape dataset: distribution shift detection results

| Method | InD Dataset | InD-Anomalies | | | Woodscape / Cityscapes | | | Woodscape-Soiling | | |
|---|---|---|---|---|---|---|---|---|---|---|
| | | FPR95 ↓ | AUROC ↑ | AUPR ↑ | FPR95 ↓ | AUROC ↑ | AUPR ↑ | FPR95 ↓ | AUROC ↑ | AUPR ↑ |
| LaRED-PCA50 | Cityscapes | 31.21 | 90.88 | 89.43 | 14.23 | 9.71 | 97.37 | 7.94 | 97.88 | 98.25 |
| LaRED-PCA50 | Woodscapes | 17.4 | 97.28 | 97.87 | 7.11 | 98.24 | 98.7 | 35.94 | 90.73 | 91.31 |

Table 20: U-Net distribution shift detection results

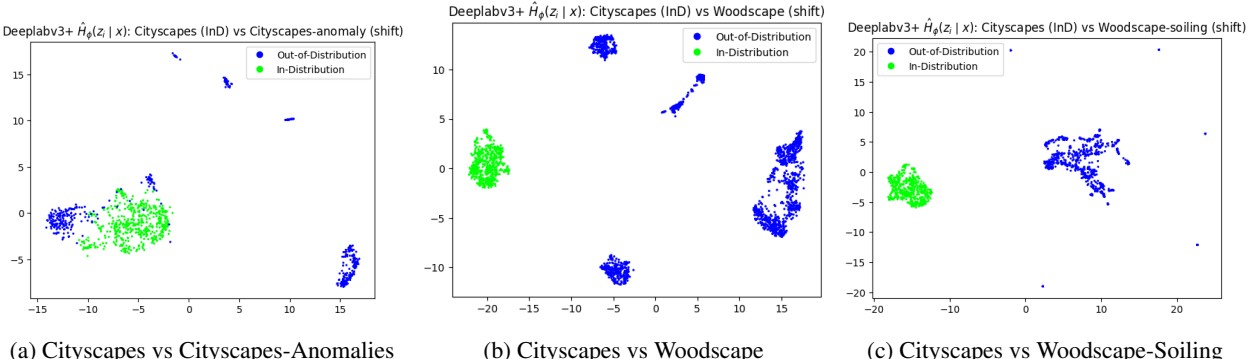

(a) Cityscapes vs Cityscapes-Anomalies    (b) Cityscapes vs Woodscape    (c) Cityscapes vs Woodscape-Soiling

Figure 13: Deeplabv3+ Cityscapes (InD): Entropy vectors 2D projection comparison using PaCMAP

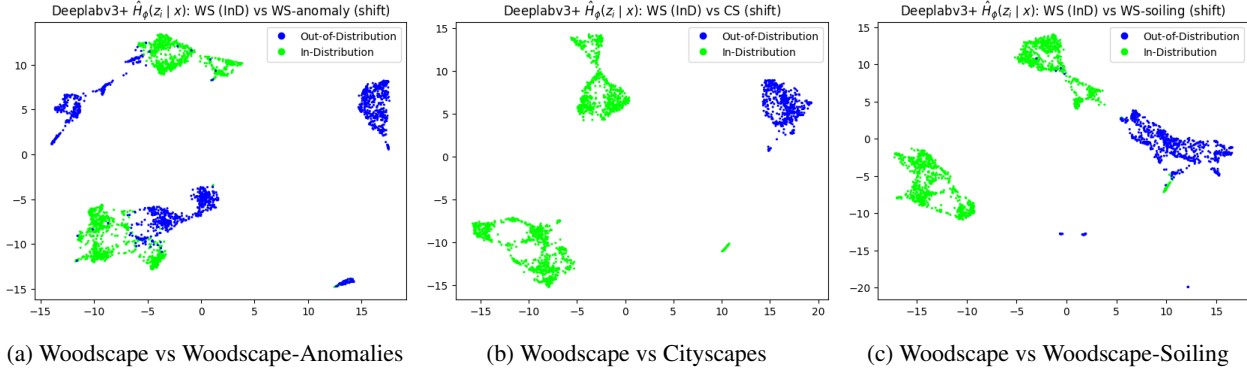

(a) Woodscape vs Woodscape-Anomalies    (b) Woodscape vs Cityscapes    (c) Woodscape vs Woodscape-Soiling

Figure 14: Deeplabv3+ trained w/Woodscape (InD): Entropy vectors 2D projection comparison using PaCMAP

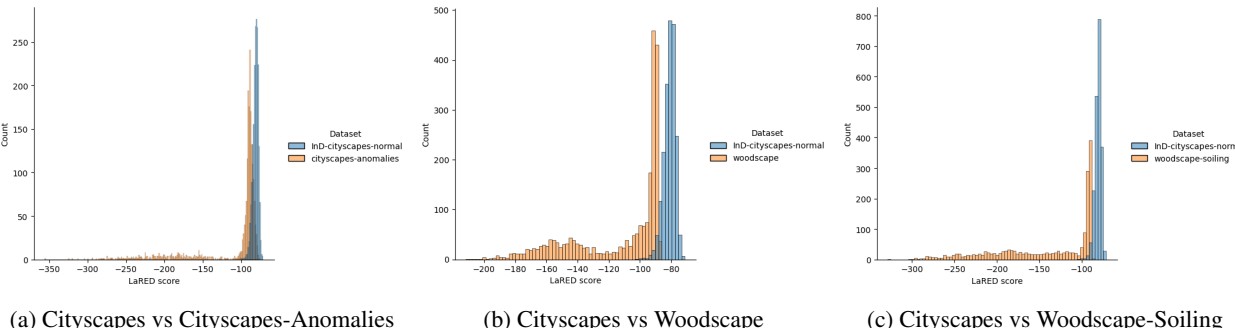

(a) Cityscapes vs Cityscapes-Anomalies  (b) Cityscapes vs Woodscape  (c) Cityscapes vs Woodscape-Soiling

Figure 15: DeepLabv3+ trained w/Cityscapes (InD): LaRED score comparison

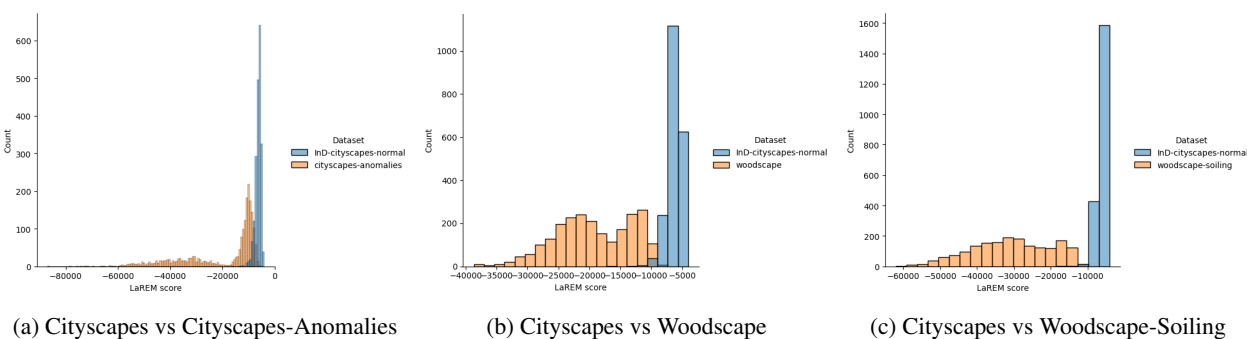

(a) Cityscapes vs Cityscapes-Anomalies  (b) Cityscapes vs Woodscape  (c) Cityscapes vs Woodscape-Soiling

Figure 16: DeepLabv3+ trained w/Cityscapes (InD): LaREM score comparison

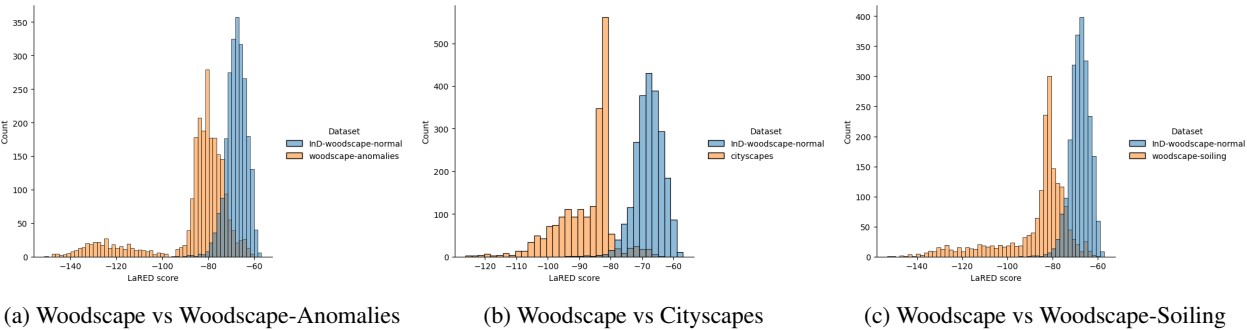

(a) Woodscape vs Woodscape-Anomalies  (b) Woodscape vs Cityscapes  (c) Woodscape vs Woodscape-Soiling

Figure 17: DeepLabv3+ trained w/Woodscape (InD): LaRED score comparison

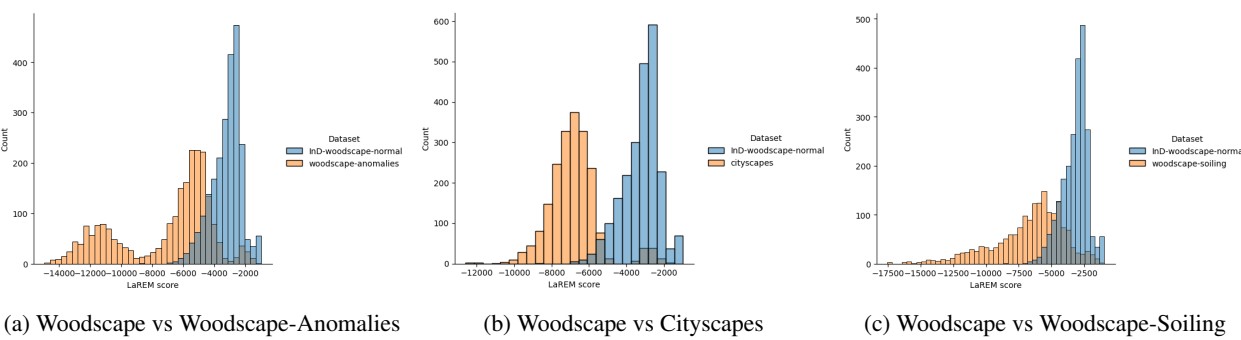

(a) Woodscape vs Woodscape-Anomalies  (b) Woodscape vs Cityscapes  (c) Woodscape vs Woodscape-Soiling

Figure 18: DeepLabv3+ trained w/Woodscape (InD): LaREM score comparison

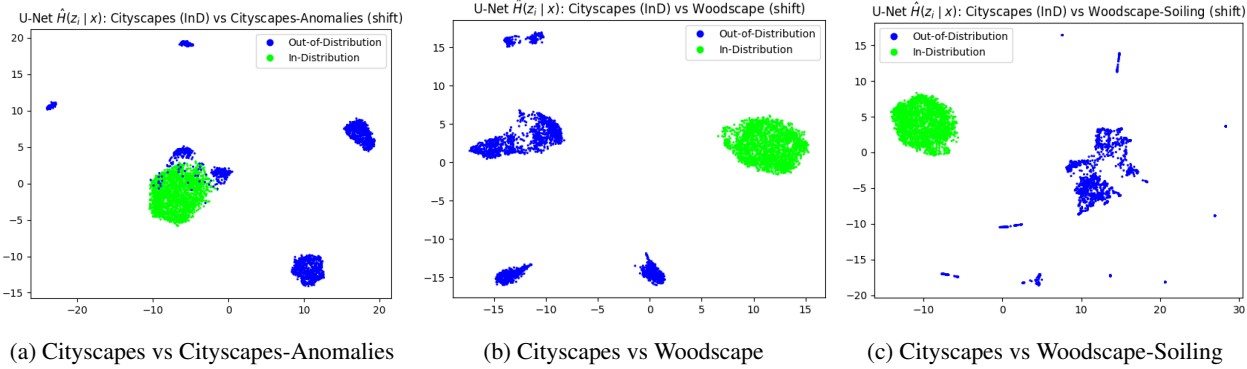

(a) Cityscapes vs Cityscapes-Anomalies     (b) Cityscapes vs Woodscape     (c) Cityscapes vs Woodscape-Soiling

Figure 19: U-Net trained w/Cityscapes (InD): Entropy vectors 2D projection comparison using PaCMAP

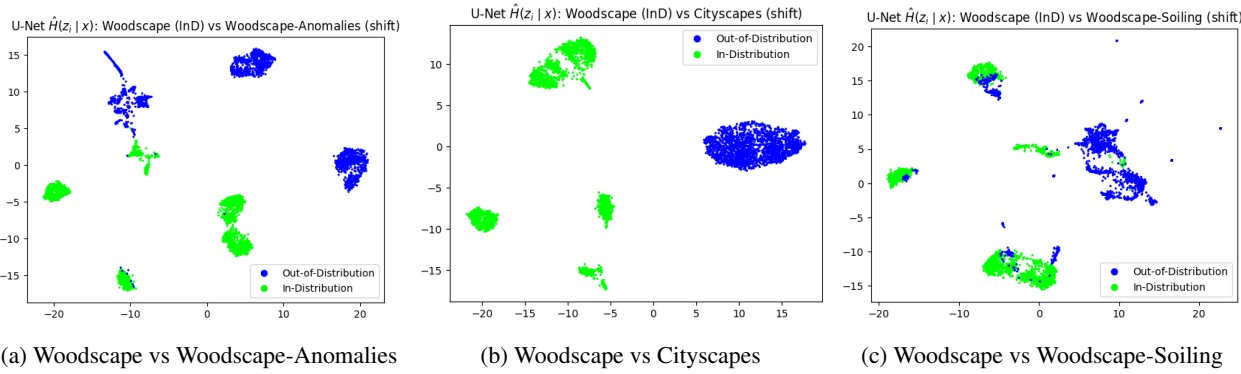

(a) Woodscape vs Woodscape-Anomalies     (b) Woodscape vs Cityscapes     (c) Woodscape vs Woodscape-Soiling

Figure 20: U-Net trained w/Woodscape (InD): Entropy vectors 2D projection comparison using PaCMAP

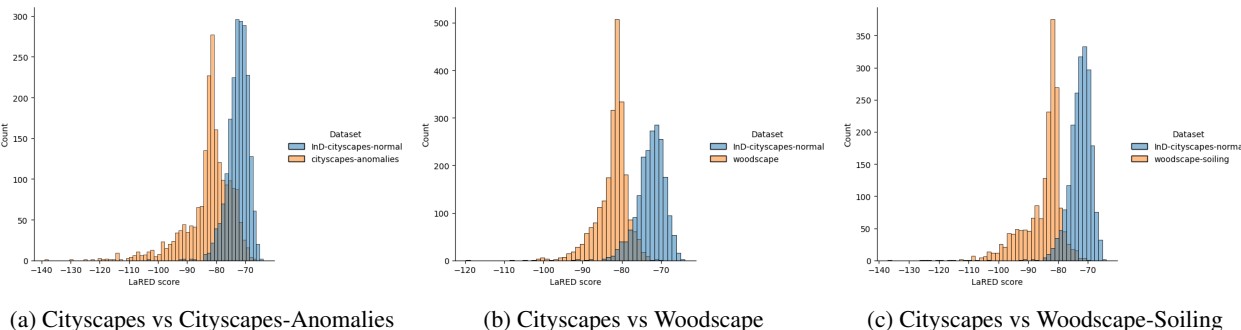

(a) Cityscapes vs Cityscapes-Anomalies     (b) Cityscapes vs Woodscape     (c) Cityscapes vs Woodscape-Soiling

Figure 21: U-Net trained w/Cityscapes (InD): LaRED score comparison

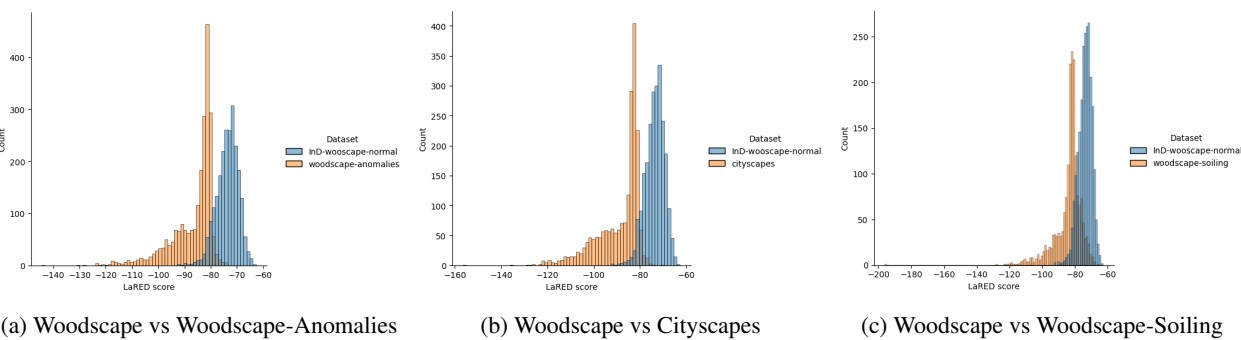

(a) Woodscape vs Woodscape-Anomalies     (b) Woodscape vs Cityscapes     (c) Woodscape vs Woodscape-Soiling

Figure 22: U-Net trained w/Woodscape (InD): LaRED score comparison

# F  IMAGE LEVEL DETECTION VS MORE DETAILED DETECTION SCHEMES

We performed image-level OoD detection, where the whole input image is classified as InD or OoD, even for object detection and semantic segmentation tasks. In this regard, our results allow us to wonder if more detailed or localized detection schemes alone are sufficient for detecting distribution shifts in complex computer vision tasks. In object detection, our results from Table 2 show that adapted simple post-hoc methods for image level detection can surpass recent *SotA* object level detection methods [Du et al., 2022, Wilson et al., 2023]. In semantic segmentation, recent benchmarks [Chan et al., 2021] also consider adapted post-hoc methods for anomaly detection at the pixel level. Nevertheless, the execution runtime for these methods is prohibitive for safety-critical applications with tight time constraints. Therefore, we believe that image-level detection can be seen as a previous or complementary step towards object-level or pixel-level OoD detection, which, for sure, are more difficult problems.

Regarding the evaluation, for the object detection task, the OoD datasets (COCO and OpenImages) are quite far away semantically and visually from the InD BDD100k. Objects, backgrounds, and scenes were all quite different, which creates an ideal situation for our proposed method and uncertainty-based confidence scores. For semantic segmentation, the evaluation was limited to covariate-shift data close to the InD datasets. In this case, the evaluation can be extended using datasets covered in anomaly segmentation benchmarks [Chan et al., 2021] and with datasets with stronger semantic shifts.

## F.1  ON SEMANTIC SEGMENTATION PREDICTIVE UNCERTAINTY WITH MCD

It is important to reveal the limitations of predictive entropy with MCD. Predictive uncertainty in semantic segmentation ends up providing a confidence measure per pixel instead of an image-level confidence measure. A qualitative inspection of predictive entropy maps in Figure 23 shows no noticeable difference in the predictions from semantically similar samples. Concretely, a DeepLabv3+ DNN trained with the Woodscape dataset is sufficiently robust to handle input samples from the Cityscapes. Although robustness is a desired property in DNNs, we cannot assume that the validation or test set performance will hold for new "shifted" samples. Moreover, image perturbation due to environmental factors can lead to wrong overconfident predictions, as illustrated in Figure 24. From a strict safety point of view, it is impossible to provide performance guarantees given the high dimensional input space and the ignorance of all the potential factors that can cause or lead to a data distribution shift. Safety is about rare, high-consequence events as those depicted in  Figure 24. Therefore, the detection of both mild and drastic distribution shifts is paramount for safe deployment and to elicit trust in the DNN-based component, as shown with both of our proposed confidence scores `LaRED` & `LaREM` in Appendix E.

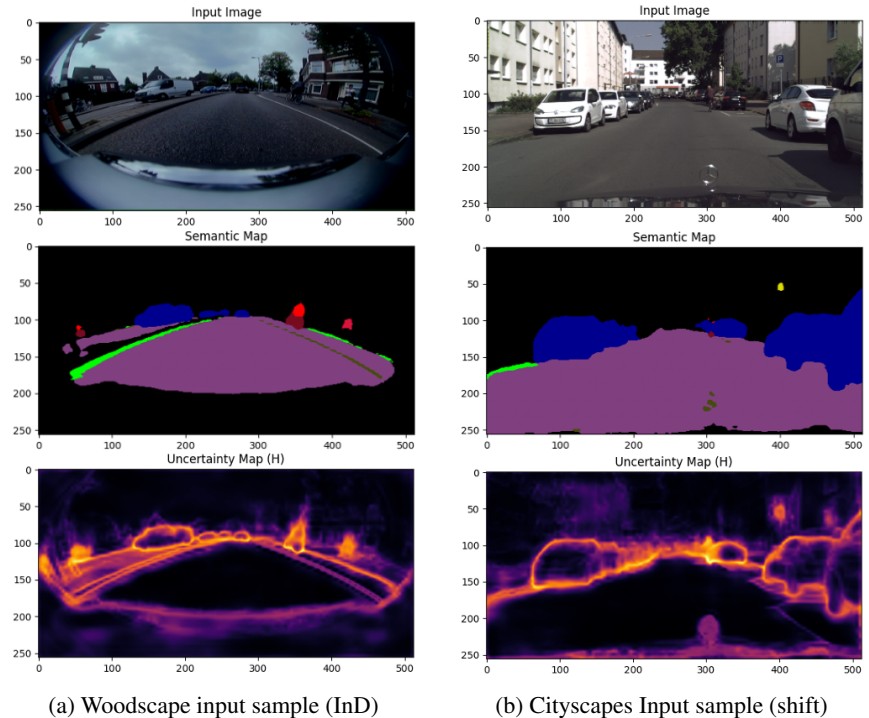

(a) Woodscape input sample (InD)   (b) Cityscapes Input sample (shift)

Figure 23: DeepLabv3+ MCD predictions for an InD sample vs Cityscapes (shift) sample, denoting the DNN robustness

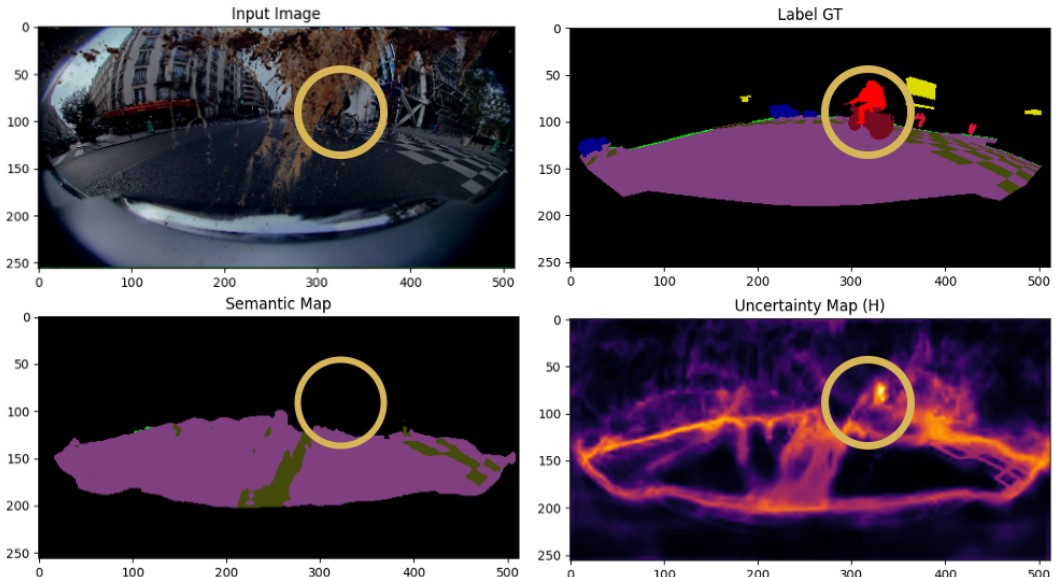

Figure 24: Deeplabv3+ MCD predictions and predictive entropy qualitative comparison for InD sample w/Covariate shift. The top row shows the input image with mud on lens perturbation and the ground truth labels. The bottom row shows the DNN MCD predicted semantic and entropy maps. The yellow circle highlights the wrong overconfident predictions when a relevant actor in the environment is partially occluded by the mud perturbation, exhibiting the DNN's lack of understanding of semantic structures and contexts