# OpenReview forum: "Latent Representation Entropy Density for Distribution Shift Detection"
_auai.org/UAI/2024/Conference — UAI 2024 poster_

### Official Review · Reviewer_C9Pu · 2024-03-04

**Q2-1 Originality-Novelty:** 2
**Q2-2 Correctness-Technical Quality:** 3
**Q2-5 Clarity Of Writing:** 2

**Q1 Summary And Contributions:**

The paper proposes a novel method for detecting Out-of-Distribution (OOD) samples from latent representations.

---

Postal Rebuttal: The author adequately addressed my questions. I have raised my score from 5 to 6.

**Q2-3 Extent To Which Claims Are Supported By Evidence:**

3: Good: the main claims are supported by convincing evidence (in the form of adequate experimental evaluation, proofs, (pseudo-)code, references, assumptions).

**Q2-4 Reproducibility:**

2: Fair: key resources (e.g. proofs, code, data) are unavailable but key details (e.g. proof sketches, experimental setup) are sufficiently well-described for an expert to confidently reproduce the main results.

**Q3 Main Strengths:**

The paper conducted a systematic evaluation of OOD detection across three computer vision tasks: classification, detection, and segmentation.

The proposed methods are comparable to state-of-the-art methods.

**Q4 Main Weakness:**

The presentation of the core results lacks clarity. The author includes numerous tables and figures without conveying the main messages effectively. For instance, regarding the results in Table 1, the author mentions, "We believe that the drop in performance is due to sub-optimal parameter selection, i.e., the parameters of the methods are sensitive to the OoD evaluation dataset." This explanation does not appear to be a valid reason. It raises the question of why optimal parameters for those baseline methods could not be identified.

**Q5 Detailed Comments To The Authors:**

- Why is the KNN baseline included in classification and segmentation but not in detection?
- Why is the MSP baseline included in classification and detection but not in segmentation?
- Why does the Mahalanobis method perform well in segmentation but poorly in classification?

**Q9 Complying With Reviewing Instructions:**

Yes

---

> ### Author Rebuttal · Authors · 2024-04-05
>
> Dear reviewer,
>
> We thank you for taking the time to read and carefully dig into the details of our proposed method. We appreciate that you recognized that our method archives comparable results to SotA methods across three computer vision tasks: classification, object detection, and semantic segmentation.
>
> We hope to clarify your concerns below:
>
> > "We believe that the drop in performance is due to sub-optimal parameter selection, i.e., the parameters of the methods are sensitive to the OoD evaluation dataset." This explanation does not appear to be a valid reason. It raises the question of why optimal parameters for those baseline methods could not be identified.
>
> Indeed, the hyperparameter selection for the baselines might not have been clearly described in paragraph "Results" from Section 4.1. **We used the hyperparameters that each baseline method/paper suggests**. More details on this aspect are found in paragraph "Baseline Methods" from section 4.2, where we present the hyperparameters for the ReAct [1], ASH [2], and DICE [3] baselines. However, it is out of the scope of our work to search if there are better hyperparameters for each of the baselines and evaluation benchmarks in each task.
>
>
> > "Why is the KNN baseline included in classification and segmentation but not in detection?"
>
> Thanks for pointing this out! We prioritized output-based baseline methods in the object detection task due to space and time constraints. Recall that all the baselines presented in Table 2 were implemented at the image level, not at the object level, to make a fair comparison with our method. In addition, we reported the results from VOS and SAFE papers to contrast the detection performance between image-level and object-level detection.
>
> Below, you can find the results of the kNN baseline on the object detection task and the corresponding DNN:
>
> |   **InD BDD-100K**  | **OoD-COCO** |            | **OoD-OpenImages** |            |
> |:-------------------:|:-------------:|------------|:-------------------:|------------|
> |                     |   **FPR95**   |  **AUROC** |      **FPR95**      | **AUROC**  |
> | LaREM RPN img-lvl (ours)       | 0.74±0.42    | 99.77±0.29 | 0.10±0.08          | 99.91±0.08 |
> | kNN img-lvl        | 26.59±1.13    | 93.05±1.63 | 19.36±1.47          | 94.94±1.82 |
> | kNN obj-lvl (SAFE) | 47.28         | 87.45      | 44.50               | 88.37      |
>
> The results above also show the kNN baseline results reported in SAFE [6], contrasting the image-level and object-level detection performance.
>
>
> > Why is the MSP baseline included in classification and detection but not in segmentation?
>
> MSP is an output-based method; therefore, in the case of semantic segmentation (sem-seg),  it provides a confidence score per pixel (in the predicted semantic mask image). For this reason, a direct comparison is not possible. In addition, in sem-seg, the inference/computation time of the MSP score is very high, and this is also the case for other output-based scores adapted from the image classification task. We refer to [5] Section D.4 and Table 5  from the appendix for a comparison of output-based scores runtime in the sem-seg task.
>
>
> > Why does the Mahalanobis method perform well in segmentation but poorly in classification?
>
> This is a very interesting question! A plausible reason for the results of this baseline is the information content in the feature maps. In the classification case (ResNet18), the dimension of the penultimate representation is 512, while in the penultimate of the Deeplabv3+ architecture (decoder block), the number of feature maps is 256. However, the size of feature maps in the Deeplabv3+ is bigger than that of the ResNet-18.
>
> Moreover, we noticed that distance-based measures (e.g., Mahalanobis) perform well in the far OoD setting (semantic shift + covariate shift) and worse in near OoD. Also, from the results in Section D from the supp. material we see that LaREM works better than LaRED at higher dimensionality (under the same conditions: DNN, location, feature map size,...), which might also be associated with your observation.
>
> Nonetheless, one general insight from our results (see section 4.5) and other works such as [4] is that no single technique seems to have the best performance for all benchmarks. This means that some methods work better on some architectures and datasets and worse on others, analog to a *no-free lunch* for OoD detection.
>
> Please let us know if these answers address your concerns/questions and if you would consider updating your score.
>
> **References**
>
> [1] https://arxiv.org/pdf/2111.12797.pdf
>
> [2] https://arxiv.org/pdf/2209.09858.pdf
>
> [3] https://arxiv.org/pdf/2111.09805.pdf
>
> [4] https://arxiv.org/pdf/2306.09301.pdf
>
> [5] https://arxiv.org/pdf/2104.14812.pdf
>
> [6] https://openaccess.thecvf.com/content/ICCV2023/papers/Wilson_SAFE_Sensitivity-Aware_Features_for_Out-of-Distribution_Object_Detection_ICCV_2023_paper.pdf

---

### Official Review · Reviewer_Y8hV · 2024-03-04

**Q2-1 Originality-Novelty:** 2
**Q2-2 Correctness-Technical Quality:** 1
**Q2-5 Clarity Of Writing:** 2

**Q1 Summary And Contributions:**

This manuscript introduces two confidence scores for detecting distribution shift by estimating uncertainty or entropy in the latent representations. It is reasonable to utilize entropy estimated from latent representations to identify distribution shift.

**Q2-3 Extent To Which Claims Are Supported By Evidence:**

3: Good: the main claims are supported by convincing evidence (in the form of adequate experimental evaluation, proofs, (pseudo-)code, references, assumptions).

**Q2-4 Reproducibility:**

2: Fair: key resources (e.g. proofs, code, data) are unavailable but key details (e.g. proof sketches, experimental setup) are sufficiently well-described for an expert to confidently reproduce the main results.

**Q3 Main Strengths:**

The experimental results presented in this study indicate that both of the proposed measures exhibit comparable performance across multiple applications. Additionally, the authors have adequately provided the necessary materials for introducing the methodology and describing the experiments.

**Q4 Main Weakness:**

The uncertainty of a random variable is inherent in the probability distribution it follows, rather than being influenced by external factors imposed on it after the variable is determined.
Given this, although it achieved impressive results in numerous experiments for the proposed methods, I have doubts about the fundamental principles of this work, especially on the dropout operation to generate noised latent.

**Q5 Detailed Comments To The Authors:**

This manuscript has technical flaws and I suggest a Reject. If the authors can provide convincing reasons for the concerns mentioned below, I may consider changing my decision.
Regarding to uncertainty of a random variable, entropy itself is the most confident measurement, so why the dropout procedure is necessary for estimation? Furthermore, it is important to note that the uncertainty of a random variable is inherent in the probability distribution it follows, rather than being influenced by external factors imposed on it after the variable is determined. Given this, although it achieved impressive results in numerous experiments for the proposed methods, I have doubts about the fundamental principles of this work.
Additionally, the presentation should be improved and there are numerous typos and several instances of imprecise mathematical descriptions.
1. ‘’InD’’ should be explained the first time it appears, ‘Where’ after Eq.(2) should be ‘where’, and some other similar editing problems. Please check thoroughly.
2. \boldsymble{\Psi} and \psi in the first paragraph of section 3 needs more clear description, ‘\left\{ {{x_n},y{ & _n}} \right\}_n^N’ in section 3.2 should be ‘\left\{ {{x_n},y{ & _n}} \right\}_{n=1}^N’ and similar issues need to be addressed.

**Q9 Complying With Reviewing Instructions:**

Yes

---

> ### Author Rebuttal · Authors · 2024-04-04
>
> Dear reviewer,
>
> We thank you for taking the time to read and carefully dig into the details of our proposed method.
>
> > Regarding to uncertainty of a random variable, entropy itself is the most confident measurement, so why the dropout procedure is necessary for estimation?... the uncertainty of a random variable is inherent in the probability distribution it follows, rather than being influenced by external factors imposed on it after the variable is determined.
>
> Thank you for raising this concern since the intuition behind our method might not be clear. We agree that entropy is a useful measurement, and that's why we use it. We hope to clarify your concerns below:
>
> **Why do we need dropout/dropblock ?**
>
> Please note that our method obtains an entropy estimation vector per input sample. Without the noise layer, we obtain a single ‘_clean_’ representation $z$ for a given input sample $x$, which hinders the entropy estimation computation. For this reason, we rely on dropout(block) to produce a set of $M$ noisy versions of the representation $ \\{ z_{i} \\}_{i}^{M} $, which are used to estimate the representation entropy for a given input sample.
>
> Further, we can make an analogy to common data augmentation techniques in images. However, in our case, this is done at the representation level. Moreover, through the lens of contrastive learning, we can also see the noise layers as an enabler to generate positive representation samples around a clean/anchor representation for a given input sample.
>
> **The intuition behind our method**
>
> Consider the example below, where, for simplicity, we assume a Gaussian latent space (at some layer of the DNN) to use the variance (instead of directly referring to the entropy):
>
> Our 1st step is to build an uncertainty reference (a distribution):
>
> Let's consider a set of training in-distribution (InD) samples
> $
> x_{train}^{n} = \\{ x^{1}, x^{2},..., x^{n} \\}
> $
>
> Then, for each sample in the set, we obtain $M$ noisy representations (w/ the noise layer) $z^{n}=\\{ z_{i}^{n} \\}_{i}^{M}$:
>
> In this way, we can compute the variance of the representations $z^{n}$ for each input sample $x^{n}$, i.e., we get an uncertainty estimate per sample.
> $\sigma_{train}^{n} = \\{  \sigma^{1}, \sigma^{2}, …, \sigma^{n} \\}$
>
> For a new input sample $x^*$ we expect the variance (entropy) to behave differently:
>
> 1) If $x^*$ belongs to the InD family:
>
>     The variance $\sigma^*$ of the noisy representations from $x_{InD}^{*}$ will be similar to those observed in $ \sigma_{train}^{n} $
>
>     We argue that this happens due to the inherent robustness of the DNN. It’s widely known that DNNs are overparameterized models that tend to learn redundant features. Therefore, despite removing information from the feature maps (with dropout), the variability of the noisy features is similar to those observed in the training samples. Note that this description also aligns with the intuition behind the ASH method [4] (App. E and L).
>
> 2) If $x^*$ is OoD, we have two possible options:
>     - The ideal intuitive case: the variance $ \sigma_{OoD1}^* $ of the noisy representations from $ x_{OoD1}^{*} $ will be higher than those observed in $\sigma_{train}^{n}$ (under confidence).
>
>     - The counter-intutitve case: where the variance $\sigma_{OoD2}^*$ of the noisy representations from $x_{OoD2}^{*} $ will be lower than those observed in $\sigma_{train}^{n}$ (over confidence).
>
>     To explain this behavior, multiple works [1-5] have observed and argued that overconfident outputs are a result of misused in-distribution BatchNorm statistics. In this context, shifted OoD input samples trigger an abnormally large proportion of units (in a layer) and produce abnormally high activation values.
>
> We argue that dropout(block) increases the differences between the observed uncertainty for InD/OoD samples. In summary, our method seeks to leverage this abnormal activation characteristic by injecting multiplicative noise through the dropout(block) layer and also, at the same time, we take advantage of the possible redundant learned representations from overparameterized DNNs.
>
> To support our claims, Fig. 7, Figs. 13-14, Figs.19-20 from the supp. material, show the 2D projections of the entropy vectors for InD samples vs OoD samples in different computer vision tasks. In all figures, we observe well-defined clusters, which also explains the effectiveness of using latent entropy vectors as an extracted feature from the samples to perform the OoD detection task.
>
> Please let us know if these answers address your concerns and if you would consider updating your score.
>
> **References**
>
> [1] https://arxiv.org/pdf/2110.11334.pdf
>
> [2] https://arxiv.org/pdf/2111.12797.pdf
>
> [3] https://arxiv.org/pdf/2111.09805.pdf
>
> [4] https://arxiv.org/pdf/2209.09858.pdf
>
> [5] https://openaccess.thecvf.com/content/ICCV2023/papers/Wilson_SAFE_Sensitivity-Aware_Features_for_Out-of-Distribution_Object_Detection_ICCV_2023_paper.pdf

---

### Official Review · Reviewer_3Zsb · 2024-03-21

**Q2-1 Originality-Novelty:** 2
**Q2-2 Correctness-Technical Quality:** 3
**Q2-5 Clarity Of Writing:** 3

**Q1 Summary And Contributions:**

This paper propose a sample-based method for estimating uncertainty and detecting distribution shifts from intermediate representations. Specifically the authors leverage the latent representation entropy density from the training data and design two confidence scores LaRED and LaREM. The proposed method is computational faster and performs on par with state-of-the-art methods.

**Q2-3 Extent To Which Claims Are Supported By Evidence:**

3: Good: the main claims are supported by convincing evidence (in the form of adequate experimental evaluation, proofs, (pseudo-)code, references, assumptions).

**Q2-4 Reproducibility:**

3: Good: key resources (e.g. proofs, code, data) are available and key details (e.g. proofs, experimental setup) are sufficiently well-described for competent researchers to confidently reproduce the main results.

**Q3 Main Strengths:**

1. The proposed method largely close the gap between DNN methods and BDL methods. Experimental results on classification, detection, and segmentation supports the main arguments, showing that the proposed method performs on par with previous SOTAs.

**Q4 Main Weakness:**

1. Are any of the limitations brought up in the introduction (paragraph 3 in Section 1) addressed by the proposed method? Problems such as object distance and occlusion are interesting but left unaddressed throughout the paper. If the limitations are not addressed by the proposed method, then they are not suitable as motivation of the paper.
2. The proposed approach brings limited (roughly 2 times faster) improvement in runtime.
3. It would be good to present some empirical comparisons between LaRED and LaREM based on the experiments.

**Q5 Detailed Comments To The Authors:**

1. The object detection and segmentation are used for image-level distribution shift detection. How is this compared to object-level detection? Can the proposed method be applied to object-level using instance-level features?

**Q9 Complying With Reviewing Instructions:**

Yes

---

> ### Author Rebuttal · Authors · 2024-04-04
>
> Dear reviewer,
>
> We thank you for taking the time to read and carefully dig into the details of our proposed method. We appreciate that you recognized that our method archives comparable to SotA results with less computational time than other BDL alternatives, which supports our arguments.
>
> Moreover, we hope to clarify your concerns below:
>
> > "Are any of the limitations brought up in the introduction (paragraph 3 in Section 1) addressed by the proposed method?"
>
> Yes, in the third paragraph, we cite several limitations that we aim to address: in the first place, traditional BDL methods are computationally expensive. Here, we propose a computationally lighter method since we only require one forward pass through the network. Second, we achieve the performance of some post-hoc deterministic methods that have achieved SotA recently. Third, we implement this method in complex tasks such as object detection and semantic segmentation, where inherently, situations such as occlusions and object distance appear. However, as we state in our paper, the method focuses on image-level distribution shift detection. Therefore, it is out of our method's scope to address the problem of occlusion or object distance. These are problems on their own, and we use them as **examples of current limitations and motivation**. Please refer to section F of our supp. material for more on our discussion on object/pixel level vs image-level distribution shift detection. In particular, our discussion based on the observations in Fig. 23 and Fig. 24., allows us to question the reliability of the uncertainty estimates obtained w/Bayesian DL methods.
>
>
> > "The proposed approach brings limited (roughly 2 times faster) improvement in runtime"
>
> As introduced in section 4.4, and as can be seen in Table 4, most of the computational burden lies in data transfer operations between GPU-Memory-CPU. Indeed, our first interest in the method was to experimentally test it, not optimize its implementation, which can be done in future developments. Our presented implementation uses Scikit-learn functionality, which uses the CPU, whereas the DNN implementations are carried out in GPU with PyTorch. Therefore, in future work, the density estimation and inference could be implemented in GPU, and the execution time should be greatly reduced furthermore.
>
>
> > "It would be good to present some empirical comparisons between LaRED and LaREM based on the experiments."
>
> Several results are presented in performance for both LaRED and LaREM, see all tables and figures from section 4. All of them include results for both methods. For even more details, please refer to the supplementary material.
>
>
> > "The object detection and segmentation are used for image-level distribution shift detection. How is this compared to object-level detection? "
>
> That is a very interesting question! For some discussion on the image-level vs object-level distribution shift detection, please refer to section F of the supplementary material. In summary, we argue that both approaches are complementary. In some cases, OoD objects or anomalies are hardly localizable (think of a mud stain occupying a significant part of the screen or a foggy or blurry image).
>
>
> > "Can the proposed method be applied to object-level using instance-level features?"
>
> That is, again, a very interesting question! Our hypothesis is YES. In principle, we don’t see why it shouldn’t work, but of course, we would need experiments. As presented in the approach described in [1], with the use of the ROI align algorithm, it is possible to extract latent representations that correspond to bounding box predictions. For future work, indeed, we plan on extending this method to detection at the object level.
>
>
> Please let us know if these answers address your questions and if you would consider updating your score.
>
> **References**
>
> [1] https://openaccess.thecvf.com/content/ICCV2023/papers/Wilson_SAFE_Sensitivity-Aware_Features_for_Out-of-Distribution_Object_Detection_ICCV_2023_paper.pdf

---

### Official Review · Reviewer_epSd · 2024-03-23

**Q2-1 Originality-Novelty:** 3
**Q2-2 Correctness-Technical Quality:** 3
**Q2-5 Clarity Of Writing:** 3

**Q1 Summary And Contributions:**

The authors propose two uncertainty based confidence scores - LaRED and LaREM to detect distribution shifts in data. The method estimates the entropy density from intermediate representation layers rather than having to pass the input through the entire deep network. Dropout and DropBlock layers are used to estimate uncertainty. The authors demonstrate the capabilities of the two scores on variety of tasks namely as image classification, semantic segmentation and object detection.

**Q2-3 Extent To Which Claims Are Supported By Evidence:**

3: Good: the main claims are supported by convincing evidence (in the form of adequate experimental evaluation, proofs, (pseudo-)code, references, assumptions).

**Q2-4 Reproducibility:**

3: Good: key resources (e.g. proofs, code, data) are available and key details (e.g. proofs, experimental setup) are sufficiently well-described for competent researchers to confidently reproduce the main results.

**Q3 Main Strengths:**

* The score methods proposed can be computed using a single forward pass for the given input sample thus speeding up the process. The scores can be initialized with only InD data.
* Versatility of the scores is presented with detailed and extensive experiments.
* Well structured and well written

**Q4 Main Weakness:**

* Although considerable empirical results are presented, the paper lacks theoretical justification for the design.
* As pointed by the authors, the method depends on finding a suitable placement of the dropout(block) layer, which might require some experiments.

**Q5 Detailed Comments To The Authors:**

The highlight for me is the extensive experimentation and comparisons performed.
That being said, the method might require a bit more theoretical grounding to explain the motivation behind the choice of these metrics. In the introduction section, the authors point out that sample-based uncertainty confidence score methods suffer from some limitations. The proposed method seem to overcome them and perform. What is the reason behind this?
What is the reason for presenting two separate scores? Are there situations where once could expect one score to perform better than the other?

**Q9 Complying With Reviewing Instructions:**

Yes

---

> ### Author Rebuttal · Authors · 2024-04-03
>
> Dear reviewer,
>
> We thank you for highlighting the benefits and versatility of our uncertainty-based confidence scores and for considering our paper well-structured and well-written. Please see our answers to specific points below:
>
> > “Theoretical grounding to explain the motivation behind the choice of these metrics” and “what is the reason for presenting two confidence scores?”
>
> As presented in Section II of the paper (related work), we group existing methods for distribution shift/OoD detection into 3 categories: I) output-based methods, II) density-based methods, and III) distance-based methods (for a comprehensive list and categories of methods for OoD detection, see [1]). Since our proposed method and confidence scores use the intermediate representations entropy, we decided to discard output-based scores given that we DO NOT use the DNN outputs. This is different from recent works (such as ReAct [2], DICE [3], and ASH [4]) that treat and process intermediate features/representations to improve other existing output-based scores, e.g., the energy score. In this regard, the remaining density-based and distance-based score approaches are appealing and intuitive for building our own uncertainty/entropy density-based score---although not perfect, given that each family of methods has limitations [1, 5, 6].
>
>
>
> > Paper weakness: “the method depends on finding a suitable placement of the dropout(block) layer, which might require some experiments”
>
> We agree with this observation. However, we argue that this “weakness” and the definition of other additional parameters are common in recent SotA methods [2-5] to improve the performance of the detection task, i.e., there is a trade-off.
>
>
>
> > “The authors point out that sample-based uncertainty confidence score methods suffer from some limitations. The proposed method seems to overcome them and perform.
> What is the reason behind this?”
>
> Multiple works [1-4, 7] have observed and argued that overconfident outputs are a result of misused in-distribution BatchNorm statistics. In this context, shifted OoD input samples trigger an abnormally large proportion of units (in a layer) and produce abnormally high activation values. When this happens, the propagation of abnormal activations throughout the DNN might cause high logit values, resulting in wrong predictions that are overconfident. This affects not only DNN predictive or output uncertainty-based scores (e.g., with Bayesian DL methods) but also all output-based confidence scores (MSP, energy,...). In contrast, in our method, we seek to leverage this abnormal activation characteristic by injecting multiplicative noise through the dropout(block) layer and also by taking advantage of the possible redundant learned features/representations from overparameterized DNNs.
>
> To illustrate the before-mentioned problem, in section F.1 of the paper Appendix, we show the predictive entropy (obtained w/Monte Carlo dropout) under input samples with mild and strong covariate shifts for the semantic segmentation task. Moreover, in semantic segmentation tasks, vanilla loss functions do not encourage learning semantic structures and contexts for the predictive semantic mask, potentially aggravating the problem mentioned above. The observations from Fig. 23 and Fig. 24 made us question the reliability of predictive/output uncertainty and also motivated our search for more “useful” uncertainty at different places of the DNN.
>
>
>
> > Are there situations where one could expect one score to perform better than the other?
>
> This is an interesting question! And YES, in particular, the score performance depends on the dimensionality of the entropy vectors. As mentioned at the end of section 3.2, the LaRED score uses KDE for estimating the entropy density. Therefore, the limitations of KDE that arise from the curse of dimensionality (computational stability and difficulties in parameter selection) have an impact on the performance of the LaRED score, as described in sections D.2 and D.3 and shown in Fig.6  and Table 15 from the Appendix.
>
>
> Please let us know if these address your concerns and if you would consider updating your score.
>
> > References:
> - [1] https://arxiv.org/pdf/2110.11334.pdf
> - [2] https://arxiv.org/pdf/2111.12797.pdf
> - [3] https://arxiv.org/pdf/2111.09805.pdf
> - [4] https://arxiv.org/pdf/2209.09858.pdf
> - [5] https://arxiv.org/pdf/2204.06507.pdf
> - [6] https://openaccess.thecvf.com/content/ICCV2023/papers/Park_Nearest_Neighbor_Guidance_for_Out-of-Distribution_Detection_ICCV_2023_paper.pdf
> - [7] https://openaccess.thecvf.com/content/ICCV2023/papers/Wilson_SAFE_Sensitivity-Aware_Features_for_Out-of-Distribution_Object_Detection_ICCV_2023_paper.pdf

---

### Official Review · Reviewer_Qgvt · 2024-03-23

**Q2-1 Originality-Novelty:** 3
**Q2-2 Correctness-Technical Quality:** 4
**Q2-5 Clarity Of Writing:** 4

**Q1 Summary And Contributions:**

This paper focuses on the problem of detecting distribution shift within input features. Essentially, the authors proposed two confidence scoring models, LaREM and LaRED, for detecting data distribution shifts at the image level, which rely on deep neural networks (DNNs). This study is built on many other previous works, including ideas like entropy estimation utilizing kernel density estimation or density Mahalanobis distance with normal distribution, Monte-Carlo dropout.
Their proposed method is demonstrated to be effective and computationally faster than existing SotA methods, making it a practical alternative for uncertainty-based confidence scores. An extensive experimentation has been done against standard baselines and benchmarks.

**Q2-3 Extent To Which Claims Are Supported By Evidence:**

4: Excellent: all claims are supported by very convincing evidence (in the form of comprehensive experimental evaluation, rigorous mathematical proofs, detailed (pseudo-)code, precise references, well-motivated and realistic assumptions) and the authors deliver what they promise.

**Q2-4 Reproducibility:**

4: Excellent: key resources (e.g. proofs, code, data) are available and key details (e.g. proof sketches, experimental setup) are comprehensively described for competent researchers to confidently and easily reproduce the main results.

**Q3 Main Strengths:**

It is an excellent paper, and all the claims are supported by very convincing evidence and further theoretical discussions. The paper is well-organized and clearly written. The topic perfectly aligns with the conference theme, and the paper is likely to have a constructive and significant impact on the field.
The provided code base helps in understanding the proposed method and for reproducibility.

**Q4 Main Weakness:**

I could not find a significant weakness in the paper.

**Q5 Detailed Comments To The Authors:**

Based on Figure 1, it seems like LaRED is utilizing KDE and its output is a multivariate normal distribution, and LaREM is using the Mahalanobis entropy density distance, and its output is assumed to be a normal distribution. Is this correct?

**Q9 Complying With Reviewing Instructions:**

Yes

---

> ### Author Rebuttal · Authors · 2024-04-03
>
> Dear Reviewer,
>
> Thank you for acknowledging our work and providing such insightful and positive feedback.
>
> Regarding the observation in Fig 1., that is correct. LaRED uses KDE for the entropy density estimation (nonparametric density), while LaREM assumes the entropy density distribution is a (multivariate) Gaussian.
>
> Please let us know if this answers your question.

---

### Meta-Review · Area_Chair_e1L7 · 2024-04-17

This paper proposes an OOD detection metric based on the entropy of latent representations.  The reviewers generally favored acceptance (1 x reject, 3 x weak accept, 1 x very strong accept), with all agreeing that the experiments support the method's effectiveness and practicality.  The primary complaint was that the theoretical justification of the method is lacking, as the draft simply states "Taking inspiration from Morningstar et al. [2021]...".  After reading the discussions and the paper itself, I find that the paper has general clarity, a novel method, and strong empirical results.  However, for revisions, I urge the authors to at least provide background on typical sets [Nalisnick et al., 2019] and density of states estimation [Morningstar et al, 2021], as this will help with some of the motivation / justification issues identified by reviewers and is work that is more related than most of what is discussed in the Related Work section.


Nalisnick, Eric, et al. "Detecting out-of-distribution inputs to deep generative models using typicality." arXiv preprint arXiv:1906.02994 (2019).

Morningstar, Warren, et al. "Density of states estimation for out of distribution detection." International Conference on Artificial Intelligence and Statistics. PMLR, 2021